# ASXLs binding to the PHD2/3 fingers of MLL4 provides a mechanism for the recruitment of BAP1 to active enhancers

Yi Zhang[1,8,9], Guojia Xie [2,9], Ji-Eun Lee [2,9], Mohamad Zandian [1,9], Deepthi Sudarshan[3,9], Benjamin Estavoyer[4], Caroline Benz [5], Tiina Viita[3], Golareh Asgaritarghi[3], Catherine Lachance[3], Clémence Messmer[4], Leandro Simonetti [5], Vikrant Kumar Sinha[1], Jean-Philippe Lambert [3], Yu-Wen Chen[6], Shu-Ping Wang [6], Ylva Ivarsson [5], El Bachir Affar [4,7], Jacques Côté [3] ✉, Kai Ge[2] ✉ & Tatiana G. Kutateladze [1] ✉

The human methyltransferase and transcriptional coactivator MLL4 and its paralog MLL3 are frequently mutated in cancer. MLL4 and MLL3 mono-methylate histone H3K4 and contain a set of uncharacterized PHD fingers. Here, we report a novel function of the PHD2 and PHD3 (PHD2/3) fingers of MLL4 and MLL3 that bind to ASXL2, a component of the Polycomb repressive H2AK119 deubiquitinase (PR-DUB) complex. The structure of MLL4 PHD2/3 in complex with the MLL-binding helix (MBH) of ASXL2 and mutational analyses reveal the molecular mechanism which is conserved in homologous ASXL1 and ASXL3. The native interaction of the Trithorax MLL3/4 complexes with the PR-DUB complex in vivo depends solely on MBH of ASXL1/2, coupling the two histone modifying activities. ChIP-seq analysis in embryonic stem cells demonstrates that MBH of ASXL1/2 is required for the deubiquitinase BAP1 recruitment to MLL4-bound active enhancers. Our findings suggest an ASXL1/2-dependent functional link between the MLL3/4 and PR-DUB complexes.

Mixed lineage leukemia 3 and 4 (MLL3 and MLL4, or KMT2C and KMT2D) mediate embryonic development, cell differentiation and hematopoiesis[1,2]. *MLL3* and *MLL4* are recurrently mutated in cancer patients and recognized among the top ten cancer driver genes[3,4]. These major human methyltransferases mono-methylate lysine 4 of histone H3, producing H3K4me1, a posttranslational modification (PTM) enriched in transcriptional enhancer regions and associated with active expression of tissue-specific genes[5–7]. Like all KMT2 family members, MLL3 and MLL4 form large multi-subunit complexes. Along with the core subunits WDR5, ASH2L, RBBP5, and DPY30, common to all KMT2 complexes, MLL3/4 complexes contain unique accessory subunits, such as a co-activator of nuclear receptors NCOA6, co-regulators PA1 and PTIP, and the H3K27 demethylase UTX/KDM6A[8,9]. These accessory subunits enable distinct functions of the MLL3/4 complexes, facilitate their loci-specific recruitment based on cell type and stage of differentiation, and stimulate p300/CBP-dependent acetylation of H3K27 to activate transcription[10–12]. MLL3 and MLL4 are large, 4911-residue and 5537-residue, respectively, paralogous

[1]Department of Pharmacology, University of Colorado School of Medicine, Aurora, CO 80045, USA. [2]Laboratory of Endocrinology and Receptor Biology, National Institute of Diabetes and Digestive and Kidney Diseases, NIH, Bethesda, MD 20892, USA. [3]St-Patrick Research Group in Basic Oncology, Oncology Division of CHU de Québec-Université Laval Research, Laval University Cancer Research Center, Quebec City, QC G1R 3S3, Canada. [4]Maisonneuve-Rosemont Hospital Research Center, Montréal, QC H1T 2M4, Canada. [5]Department of Chemistry, BMC, Uppsala University, Uppsala 75237, Sweden. [6]Institute of Biomedical Sciences, Academia Sinica, Taipei 11529, Taiwan, ROC. [7]Department of Medicine, University of Montréal, Montréal, QC H3C 3J7, Canada. [8]Present address: Department of Biochemistry, Case Western Reserve University, Cleveland, OH 44106, USA. [9]These authors contributed equally: Yi Zhang, Guojia Xie, Ji-Eun Lee, Mohamad Zandian, Deepthi Sudarshan. ✉e-mail: jacques.cote@crchudequebec.ulaval.ca; kai.ge@nih.gov; tatiana.kutateladze@cuanschutz.edu

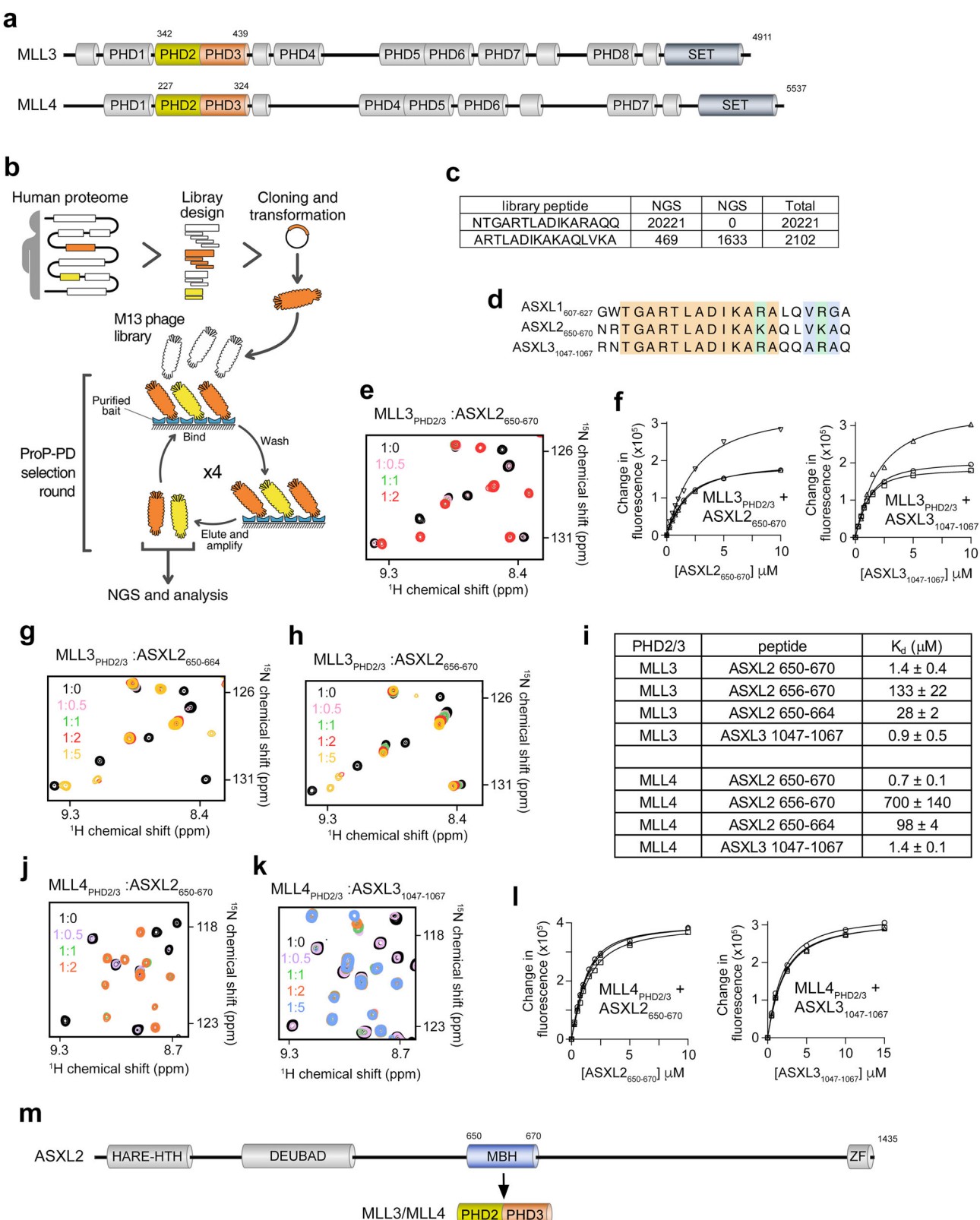

enzymes. They share overall similar domain architecture, harboring the catalytic su(var)3-9, enhancer of zeste, trithorax (SET) domain, a pair of the FY-rich (FYRN and FYRC) domains, and a set of plant homeodomain (PHD) fingers, eight in MLL3 and seven in MLL4[13] (Fig. 1a). Although the structure-function activities for the majority of these PHD fingers remain unknown, the sixth PHD (PHD6) finger of MLL4 and the homologous seventh PHD (PHD7) finger of MLL3 were

identified as readers of acetylated lysine 16 of histone H4[14,15], and the second and third PHD (PHD2/3) fingers of MLL3 were shown to mediate association of MLL3 with the histone H2A deubiquitinase and tumor suppressor BAP1 (BRCA-1 associated protein 1)[16].

MLL3 and MLL4 belong to the Trithorax group proteins that act as positive transcriptional co-regulators of developmental genes during embryonic development, and this activity is counterbalanced by the

**Fig. 1 | MLL3$_{PHD2/3}$ and MLL4$_{PHD2/3}$ interact with ASXLs. a** MLL3 and MLL4 (KMT2C and KMT2D) lysine methyltransferases domain architecture. Sequence alignment of all PHD fingers of MLL3 and MLL4 is reported in ref. 13. Schematic of the human proteome peptide-phage display library (**b**) used to identify the ASXL sequences with high confidence scores of 62% and 6.5% (**c**) as a binding partner of MLL3$_{PHD2/3}$. **d** Alignment of amino acid sequences of ASXL1/2/3. **e** Superimposed $^1$H,$^{15}$N HSQC spectra of MLL3$_{PHD2/3}$ collected upon titration with ASXL2 (aa 650-670 of ASXL2) peptide. Spectra are color coded according to the protein:peptide molar ratio. **f** Binding curves used to determine $K_d$ values by fluorescence spectroscopy. **g, h** Superimposed $^1$H,$^{15}$N HSQC spectra of MLL3$_{PHD2/3}$ collected upon titration with ASXL2 (aa 650-664) and ASXL2 (aa 656-670) peptides. Spectra are color coded

according to the protein:peptide molar ratio. **i** Binding affinities of MLL3$_{PHD2/3}$ and MLL4$_{PHD2/3}$ for the indicated ASXL2 and ASXL3 peptides measured by fluorescence spectroscopy. The $K_d$ values represent average of three independent measurements, and errors represent standard deviation. n = 3 (**j, k**) Superimposed $^1$H,$^{15}$N HSQC spectra of MLL4$_{PHD2/3}$ collected upon titration with ASXL2 (aa 650-670) and ASXL3 (aa 1047-1067) peptides. Spectra are color coded according to the protein:peptide molar ratio. **l** Binding curves used to determine binding affinities of MLL4$_{PHD2/3}$ for ASXL2 (aa 650-670) and ASXL3 (aa 1047-1067) peptides by fluorescence spectroscopy. **m** MLL3/4$_{PHD2/3}$ binds to MBH (MLL-binding helix) of ASXL2. Domain architecture of ASXL2 with MBH colored blue. MLL3/4$_{PHD2/3}$ is depicted as in (**a**).

Polycomb group proteins that maintain repressive state of the developmental genes[17]. The additional sexcombs like (ASXL) group of genes is involved in the maintenance of both transcription and repression[18–20]. Three homologous human ASXL proteins, ASXL1, ASXL2 and ASXL3, have been identified as mutually exclusive components of the Polycomb repressive deubiquitinase (PR-DUB) complex[21]. The presence of the ASXL subunit is required for activation of the catalytic subunit of this complex, BAP1, which deubiquitinates mono-ubiquitinated lysine 119 of histone H2A (H2AK119ub), a mark of repressive chromatin deposited by the Polycomb repressive complex-1 (PRC1)[22–25]. Human ASXL1 and ASXL2 are ubiquitously expressed, whereas ASXL3 expression is low and limited to a few specific tissues. ASXLs contain three domains, an uncharacterized HARE-HTH domain at the N-terminus, the DEUBAD domain which binds to BAP1 and stimulates its catalytic activity, and a C-terminal zinc finger. ASXLs and other components of the PR-DUB complex are often found mutated, truncated or misregulated in human cancers[26–28].

In this study, we identified ASXL2 and homologous ASXL1 and ASXL3 as direct binding partners of MLL3 and MLL4. We describe the structural mechanism by which the PHD2/3 fingers of MLL3/4 interact with the MLL3/4 binding helix (MBH) of ASXL2. Our in vivo data demonstrate the functional significance of this interaction and show that ASXL1/2, via MBH, links BAP1 to MLL4 on active enhancers in ESCs.

## Results and discussion

### MLL3$_{PHD2/3}$ and MLL4$_{PHD2/3}$ bind to ASXLs

We prepared a human proteome peptide-phage display library containing one million 16-residue peptides and used it to identify binding partners of the PHD2 and PHD3 fingers of MLL3 (MLL3$_{PHD2/3}$). After four rounds of selection, the enrichment of binding phages was validated through phage pool enzyme-linked immunosorbent assay (ELISA), and the peptide coding regions of binding-enriched phage pools were amplified, barcoded and sequenced (Fig. 1b). Quantitative analysis identified two overlapping peptide hits (Fig. 1c), suggesting that a highly conserved region of ASXL1, ASXL2 and ASXL3 proteins, components of the PR-DUB complex, is likely a target of MLL3$_{PHD2/3}$ (Fig. 1d). The interaction between MLL3$_{PHD2/3}$ and ASXLs was confirmed in NMR experiments. We generated $^{15}$N-labeled MLL3$_{PHD2/3}$ and recorded its $^1$H,$^{15}$N heteronuclear single quantum coherence (HSQC) spectrum. The dispersion of amide resonances in the spectrum indicated that MLL3$_{PHD2/3}$ is folded. Titration of the ASXL2 peptide (residues 650-670 of ASXL2) into the MLL3$_{PHD2/3}$ NMR sample induced substantial chemical shift perturbations (CSPs), indicative of a direct binding (Fig. 1e). These perturbations were in slow exchange regime on the NMR timescale characterized by disappearance of signals corresponding to the free state of the protein and appearance of resonances corresponding to the complex and suggested tight binding. In agreement, a 1.4 µM affinity of MLL3$_{PHD2/3}$ for the ASXL2 peptide was measured by fluorescence assays (Fig. 1f).

To define boundaries of the ASXL2 region recognized by MLL3$_{PHD2/3}$ we tested two short peptides, residues 656-670 of ASXL2 (ASXL2$_{656-670}$) and residues 650-664 of ASXL2 (ASXL2$_{650-664}$), which were six residues shorter than the original ASXL2 peptide. NMR

titrations and tryptophan fluorescence data showed that MLL3$_{PHD2/3}$ associates with the short peptides to a lesser extent compared to its association with the long ASXL2 peptide (Fig. 1g–i and Suppl. Fig. 1). The dissociation constants ($K_d$s) for the interactions of MLL3$_{PHD2/3}$ with ASXL2$_{650-664}$ and ASXL2$_{656-670}$ were found to be 28 µM and 133 µM, respectively. High sequence similarity between the ASXL2 region encompassing residues 650-670, the ASXL1 region encompassing residues 607-627, and the ASXL3 region encompassing residues 1047-1067 suggested that MLL3$_{PHD2/3}$ might also recognize ASXL1 and ASXL3 (Fig. 1d). Indeed, MLL3$_{PHD2/3}$ exhibited a 0.9 µM binding affinity toward the ASXL3 peptide in fluorescence assays and showed a similar CSPs in the slow exchange regime in NMR titration experiments with the ASXL1 peptide (Fig. 1f, i and Suppl. Fig. 2a).

The PHD2 and PHD3 fingers of homologous MLL3 and MLL4 share 70% amino acid sequence identity (Suppl. Fig. 3). We assessed whether MLL4$_{PHD2/3}$ can also interact with ASXL2, ASXL1 and ASXL3. Gradual addition of the ASXL2 peptide, or separately ASXL1 and ASXL3 peptides, to the $^{15}$N-labeled MLL4$_{PHD2/3}$ NMR sample led to large CSPs, confirming formation of the complex (Fig. 1j and Suppl. Fig. 2b). Binding of MLL4$_{PHD2/3}$ was further substantiated through measuring binding affinities for ASXL2 and ASXL3 peptides ($K_d$s = 0.7 µM and 1.4 µM, respectively) (Fig. 1i, l). Much like MLL3$_{PHD2/3}$, MLL4$_{PHD2/3}$ required the long ASXL2 peptide for strong binding and associated with the shorter ASXL2$_{650-664}$ and ASXL2$_{656-670}$ peptides weaker ($K_d$s = 98 µM and 700 µM, respectively) (Fig. 1i and Suppl. Fig. 1). Together, these results demonstrated that the recognition of the MBH (MLL binding helix, described below) region of ASXLs is a previously uncharacterized, conserved function of MLL3$_{PHD2/3}$ and MLL4$_{PHD2/3}$ and suggested that it may link the H3K4-specific methyltransferase MLL3/4 complexes with the H2AK119ub-specific deubiquitinase PR-DUB complex (Fig. 1m).

### Structural mechanism for the recognition of ASXL2 by MLL4$_{PHD2/3}$

To elucidate the molecular basis for the interaction of MLL4$_{PHD2/3}$ with ASXL2, we determined the three-dimensional structure of MLL4$_{PHD2/3}$ in complex with ASXL2 using NMR spectroscopy. We generated a linked construct containing residues 227-324 of MLL4 and residues 650-670 of ASXL2 connected by a SGPSSG linker (MLL4$_{PHD2/3}$-ASXL2). The $^1$H,$^{15}$N HSQC spectrum of the linked MLL4$_{PHD2/3}$-ASXL2 construct overlayed adequately with the spectrum of the ASXL2-bound unlinked MLL4$_{PHD2/3}$ construct, indicating formation of similar MLL4$_{PHD2/3}$:ASXL2 complexes by the unlinked and linked protein (Suppl. Fig. 4). The structure of MLL4$_{PHD2/3}$-ASXL2 shows a globular architecture of the complex with two PHD fingers being fully integrated (Fig. 2 and Suppl. Table 1). Each PHD finger folds into a double-stranded anti-parallel β-sheet and two small α-helices stabilized by two zinc ions coordinated through cysteine and histidine residues in a cross-braced topology (Fig. 2a and Suppl. Fig. 5).

In the complex, ASXL2 occupies a deep, extended binding groove at the interface of PHD2 and PHD3 and makes hydrophobic and electrostatic contacts with both modules (Fig. 2b, d). Residues R655-Q670 of ASXL2 adopt an α-helical conformation, forming an MBH, which

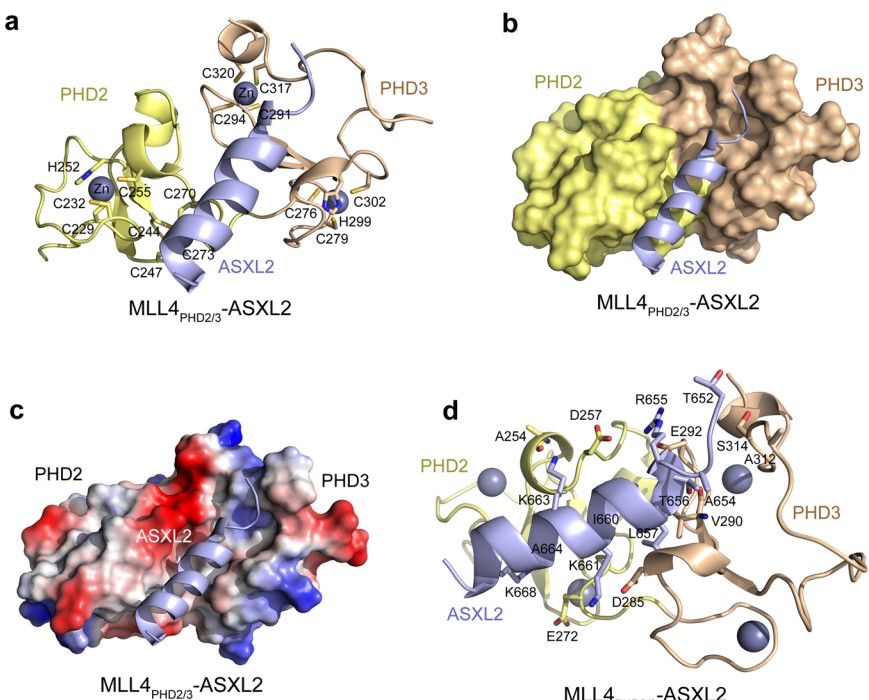

**Fig. 2 | Structure of MLL4$_{PHD2/3}$ in complex with ASXL2.** A ribbon diagram (**a**) and a surface representation (**b**) of the solution NMR structure of the PHD2 (yellow) and PHD3 (wheat) fingers of MLL4 in complex with ASXL2 (blue). Zinc ions (gray spheres) coordinating cysteine and histidine residues are shown as stick and labeled. **c** Electrostatic surface potential of the MLL4$_{PHD2/3}$:ASXL2 complex colored blue and red for positive and negative charges, respectively. **d** The MLL4$_{PHD2/3}$ and ASXL2 residues involved in the complex formation are shown in sticks and labeled.

explains the requirement for a long, ~20-residue ASXL2 peptide for robust interaction with MLL4$_{PHD2/3}$, as MBH is truncated in the shorter peptides, ASXL2$_{650-664}$ and ASXL2$_{656-670}$. Overall, the ASXL2-binding site is mainly hydrophobic in nature but is surrounded by the negatively charged walls formed by the side chains of D257, E272, D285, and E292 of MLL4$_{PHD2/3}$ (Fig. 2c, d). The hydrophobic side chains of L657, I660, and A664 of ASXL2 are fully buried in the hydrophobic pocket, whereas the side chain amino groups of K661 and K663 of ASXL2 are positioned within a hydrogen bond distance to the carboxyl side chain of D285 and the backbone carbonyl of A254 of MLL4$_{PHD2/3}$. The guanidino moiety of R655 and the hydroxyl group of T656 of ASXL2 are located within a hydrogen bond distance to the carboxyl groups of D257 and E292 and the backbone amide of V290 of MLL4$_{PHD2/3}$. Additionally, the positively charged side chains of K663 and K668 of ASXL2 are positioned to be involved in electrostatic contacts with the negatively charged side chains of D257 and E272 of MLL4$_{PHD2/3}$. The hydrophobic T652-A654 portion of ASXL2 forms a turn and makes van der Waals and hydrophobic contacts with the loop connecting the β2 strand and the C-terminal α-helix of MLL4$_{PHD2/3}$.

### Impact of the ASXL2-MLL4$_{PHD2/3}$ interface residues

To determine the role of the interface residues in the formation of the MLL4$_{PHD2/3}$:ASXL2 complex, we generated MLL4$_{PHD2/3}$ mutants and examined their structural stability and binding to ASXL2 peptide (from here on residues 650-670 of ASXL2) by NMR (Fig. 3a). All MLL4$_{PHD2/3}$ mutants but one, L256K, retained their structures, showing dispersion of amide resonances in their $^1$H,$^{15}$N HSQC spectra (Fig. 3b and Suppl. Fig. 6). A lack of CSPs in $^1$H,$^{15}$N HSQC spectra of K274E and V290K mutants of MLL4$_{PHD2/3}$ upon titration with the ASXL2 peptide indicated that K274 and V290 are required for the interaction. Substitution of D285 with lysine in MLL4$_{PHD2/3}$ led to a substantial decrease in the binding activity (K$_d$ = ~1.7 mM), and the double mutant D257K/D285K failed to associate with the ASXL2 peptide in NMR titration experiments (Fig. 3b, c and Suppl. Figs. 1, 6). The single mutations D257K and

E272K in MLL4$_{PHD2/3}$ reduced binding by ~40 fold (K$_d$s = 27 μM and 28 μM, respectively), and a ~14-fold reduction in binding activity was observed for E292K of MLL4$_{PHD2/3}$ (Fig. 3c–e and Suppl. Fig. 1). The MLL4$_{PHD2/3}$ binding affinity toward ASXL2 remained unchanged for S286A and S314A mutants (K$_d$s of 0.8 μM) (Fig. 3c and Suppl. Fig. 1).

The importance of the interfacial residues was substantiated by pull-down assays from cell lysates (Fig. 3f and Suppl. Figs. 7, 8). Resin-bound GST-MLL3$_{PHD2/3}$ and GST-MLL4$_{PHD2/3}$, wild type or mutants, were incubated with extracts from HEK293FT cells transiently transfected with full-length ASXL1 or ASXL2 (Fig. 3f). After pull downs and washes, the association of ASXL1 and ASXL2 was detected with wild type MLL3$_{PHD2/3}$ and MLL4$_{PHD2/3}$ but not with the MLL4$_{PHD2/3}$ or MLL3$_{PHD2/3}$ mutants (Fig. 3f and Suppl. Fig. 8). These results confirmed the binding of MLL3$_{PHD2/3}$ and MLL4$_{PHD2/3}$ in the context of full length ASXL1 and ASXL2 expressed in human cells. Reciprocally, mutations in MBH or the deletion of MBH in ASXL2 (ASXL2ΔMBH) also impacted this interaction (Fig. 3c, g, h and Suppl. Fig. 8). As shown in Supplementary Fig. 8, wild type MLL3$_{PHD2/3}$ and MLL4$_{PHD2/3}$ were able to pull down wild type full length ASXL2 from cell extracts, but the interaction with ASXL2ΔMBH was essentially abolished. Mutation of K661 of ASXL2, which likely forms a hydrogen bond with D285 of MLL4$_{PHD2/3}$, to a glutamate abrogated binding, and the I660E mutant of ASXL2 had a markedly decreased binding activity (K$_d$ of 1.1 mM) (Fig. 3c, g and Suppl. Fig. 1). Collectively, the mutagenesis data pointed to the critical role of both hydrophobic and electrostatic contacts in the formation of the MLL4$_{PHD2/3}$:ASXL2 complex, corroborating the structural results.

A number of mutations in the MLL4$_{PHD2/3}$ and ASXL2-binding interface have been identified in cancer patients (Cosmic). Particularly, E272K, K274E and E292K in MLL4$_{PHD2/3}$, which were tested above, are found in esophagus, breast, lung, urinary tract, and large intestine cancers. Additionally, mutations of zinc-coordinating cysteine residues, C276Y and C294R, which likely disrupt the MLL4$_{PHD2/3}$ structure, are found in large intestine cancer. The G653E mutation in ASXL2,

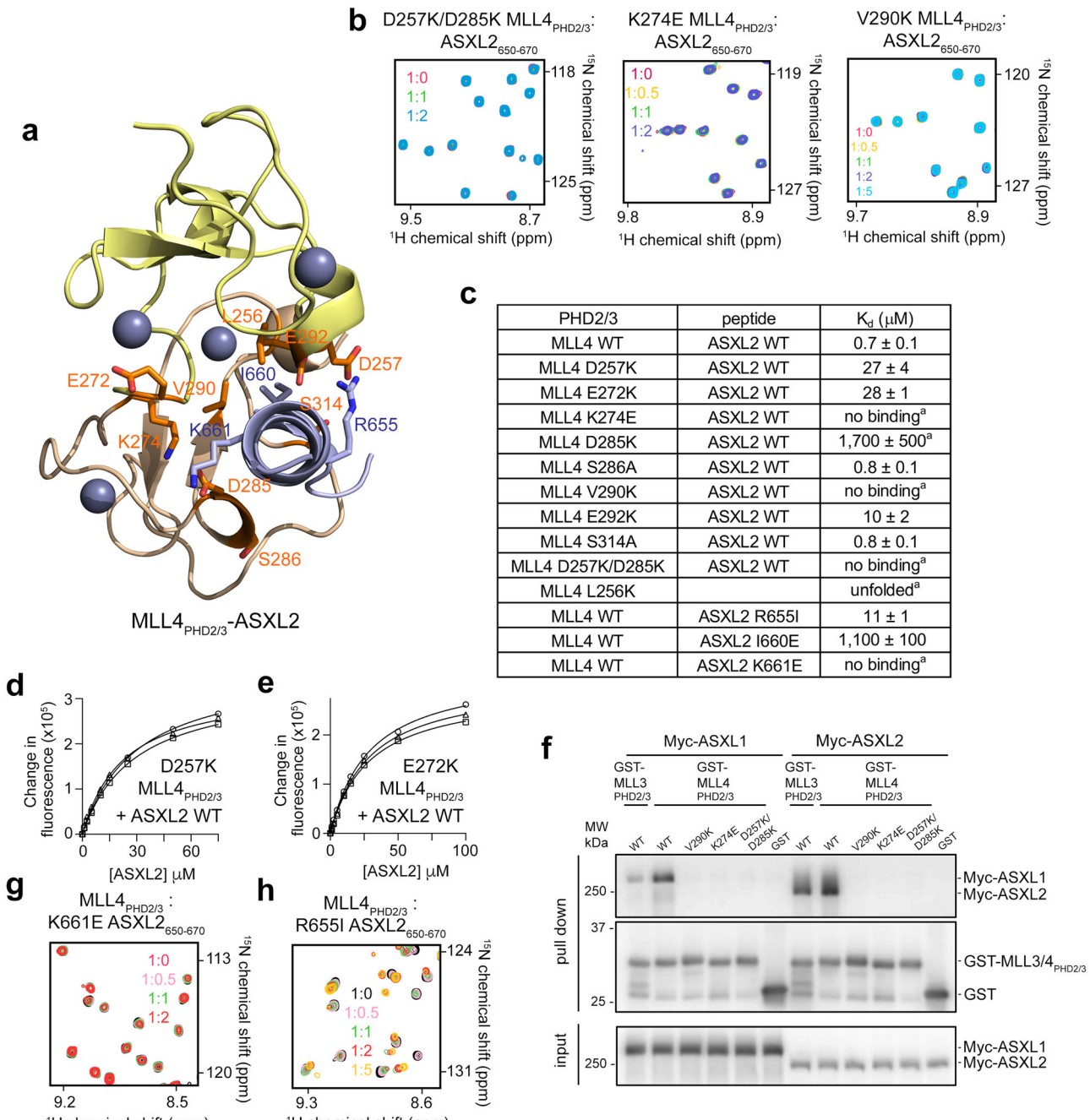

**Fig. 3 | Mutational analysis of the ASXL2-binding site. a** A ribbon diagram of the structure of the MLL4$_{PHD2/3}$:ASXL2 complex with the mutated residues shown as sticks. MLL4$_{PHD2/3}$ residues are labeled in orange and ASXL2 residues are labeled in blue. **b** Superimposed $^1$H,$^{15}$N HSQC spectra of mutated MLL4$_{PHD2/3}$ collected upon titration with the ASXL2 peptide. Spectra are color coded according to the protein:peptide molar ratio. **c** Binding affinities of WT and mutated MLL4$_{PHD2/3}$ for the indicated WT or mutated ASXL2 peptides measured by fluorescence spectroscopy or ($^a$) NMR. The $K_d$ values represent average of three independent measurements, and errors represent standard deviation. n = 3 (**d**, **e**) Binding curves used to determine binding affinities of indicated mutated MLL4$_{PHD2/3}$ for the ASXL2 peptide by fluorescence spectroscopy. **f** Western blot analysis of ASXL1 and ASXL2, pulled down by recombinant wild-type or mutated GST-MLL3$_{PHD2/3}$ and GST-MLL4$_{PHD2/3}$. GST-MLL proteins were purified from bacteria and bound to GSH resin. Human Myc-ASXL1 and Myc-ASXL2 were expressed in HEK293FT cells. n = 3 (**g**, **h**) Superimposed $^1$H,$^{15}$N HSQC spectra of WT MLL4$_{PHD2/3}$ collected upon titration with the indicated mutated ASXL2 peptides. Spectra are color coded according to the protein:peptide molar ratio.

found in central nervous system malignancies, could also affect the binding, as the small flexible glycine residue but not glutamate allows for a sharp turn of the amino-terminal part of bound ASXL2. Another mutant of ASXL2 which is associated with large intestine cancer, R655I, reduced binding of MLL4$_{PHD2/3}$ by ~16-fold (Fig. 3c, h and Suppl. Fig. 1). Overall, these findings suggest that the altered interaction between MLL4$_{PHD2/3}$ and ASXL2 could play a role in oncogenesis.

## The ASXL2-binding mechanism is conserved in MLL3$_{PHD2/3}$

Structural overlay of MLL4$_{PHD2/3}$ in complex with ASXL2 and MLL3$_{PHD2/3}$ in the apo-state (PDB ID: 2YSM) (rmsd of 1.5 Å) demonstrated that ASXL2 does not induce significant conformational changes in the PHD fingers and that the ASXL2-binding pocket is essentially pre-made (Fig. 4a). It also suggested that the molecular mechanism for the recognition of ASXL2 is conserved in MLL4 and

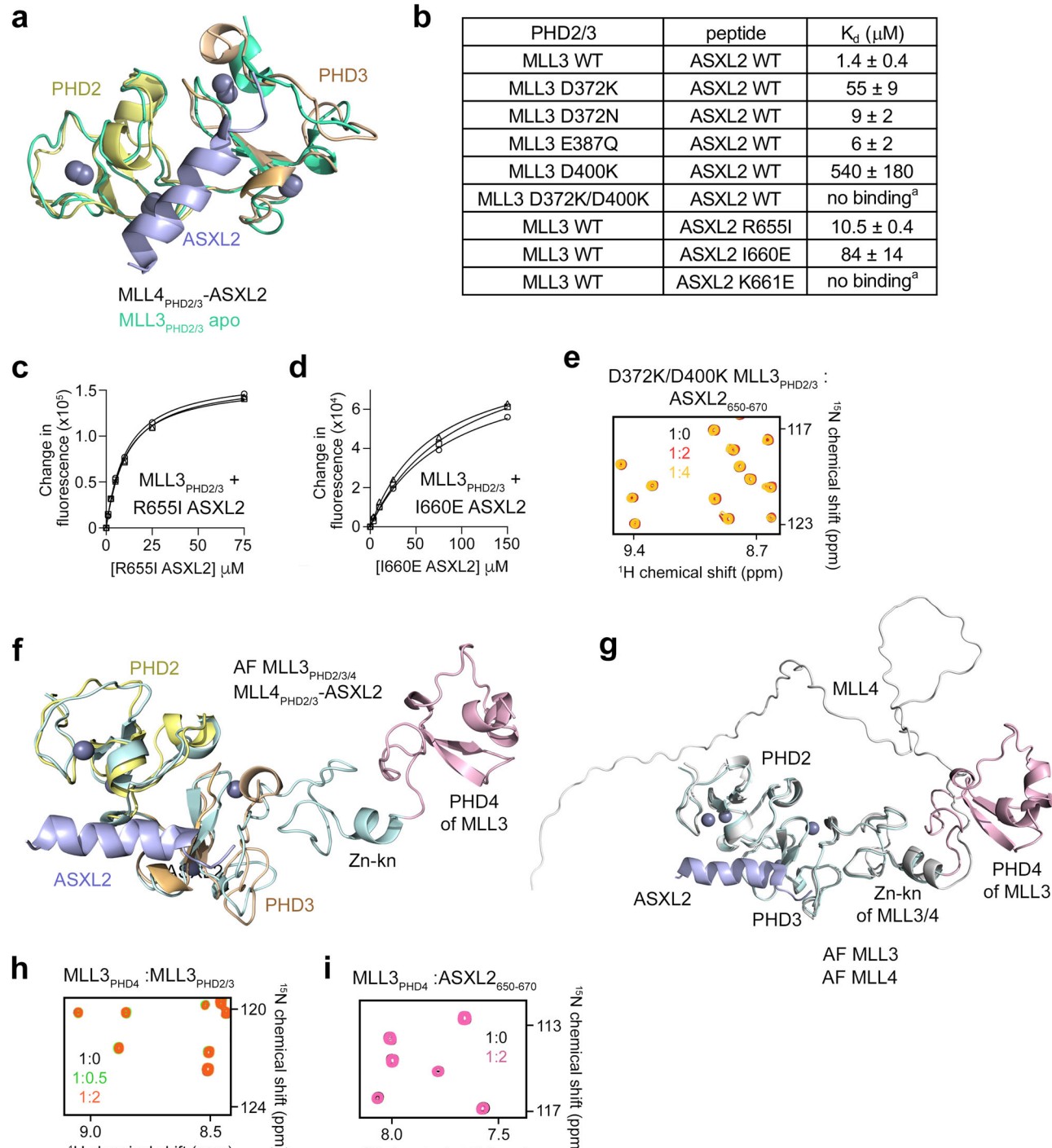

| PHD2/3 | peptide | $K_d$ (μM) |
|---|---|---|
| MLL3 WT | ASXL2 WT | 1.4 ± 0.4 |
| MLL3 D372K | ASXL2 WT | 55 ± 9 |
| MLL3 D372N | ASXL2 WT | 9 ± 2 |
| MLL3 E387Q | ASXL2 WT | 6 ± 2 |
| MLL3 D400K | ASXL2 WT | 540 ± 180 |
| MLL3 D372K/D400K | ASXL2 WT | no binding[a] |
| MLL3 WT | ASXL2 R655I | 10.5 ± 0.4 |
| MLL3 WT | ASXL2 I660E | 84 ± 14 |
| MLL3 WT | ASXL2 K661E | no binding[a] |

**Fig. 4 | Conservation of the ASXL2-binding mechanism in MLL3$_{PHD2/3}$.** **a** Overlay of the structures of the MLL4$_{PHD2/3}$:ASXL2 complex (colored as in Fig. 2) and the apo-state of MLL3$_{PHD2/3}$ (PDB ID: 2YSM) (teal). **b** Binding affinities of WT and mutated MLL3$_{PHD2/3}$ for the indicated WT and mutated ASXL2 peptides measured by fluorescence spectroscopy or ([a]) NMR. The $K_d$ values represent average of three independent measurements, and errors represent standard deviation. n = 3 (**c**, **d**) Binding curves used to determine binding affinities of WT MLL3$_{PHD2/3}$ for the indicated mutated ASXL2 peptides by fluorescence spectroscopy. **e** Superimposed $^1$H,$^{15}$N HSQC spectra of the indicated mutated MLL3$_{PHD2/3}$ collected upon titration with the ASXL2 peptide. Spectra are color coded according to the protein:peptide molar ratio. **f** Overlay of the structures of the MLL4$_{PHD2/3}$:ASXL2 complex (colored as in Fig. 2) and the AlphaFold (AF)-generated model of the apo-state of

MLL3$_{PHD2/3/4}$. The zinc-knuckle, following the PHD3 finger of MLL3, is colored cyan, and the fourth PHD finger (PHD4) of MLL3 is colored pink. **g** Overlay of the structures of the MLL4$_{PHD2/3}$:ASXL2 complex (only ASXL2 is shown and colored blue) and two AF-models. One model is the apo-state of MLL3$_{PHD2/3/4}$, with the zinc-knuckle following the PHD3 finger of MLL3 colored cyan, and PHD4 of MLL3 colored pink as in (**f**). The second model is the apo-state of MLL4 of the same length as MLL3$_{PHD2/3/4}$ (colored gray). The sequence of MLL4, corresponding to the sequence of the PHD4 finger in MLL3, is predicted to be unstructured by AF and labeled MLL4. **h**, **i** Superimposed $^1$H,$^{15}$N HSQC spectra of MLL3$_{PHD4}$ collected upon titration with the indicated ligands. Spectra are color coded according to the protein:ligand molar ratio.

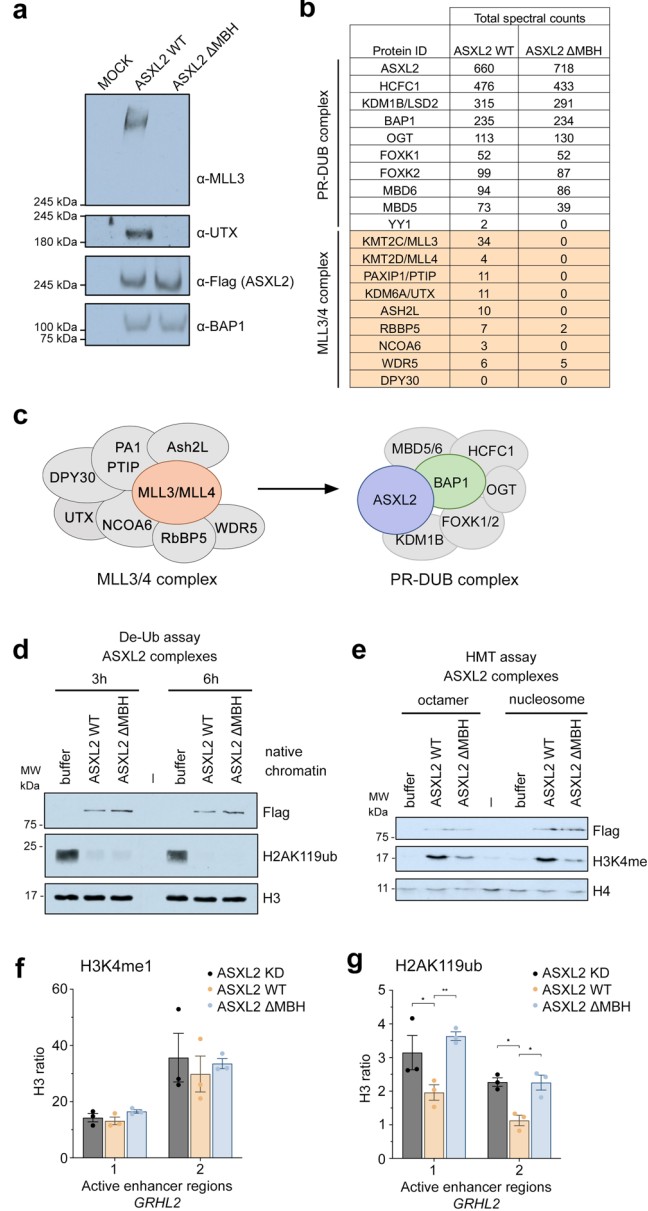

**Fig. 5 | MBH of ASXL2 directly links the native MLL3/4 and PR-DUB complexes in vivo. a** Western blot analysis of fractions obtained from the tandem affinity purification of WT ASXL2 and ASXL2 ΔMBH from K562 nuclear extracts using the indicated antibodies. Mock is the fraction from tag-only K562 cell line, subjected to the same purification protocol. (n = 1) (**b**) Proteins identified by mass spectrometry proteomic analysis of purified fractions from K562 cells expressing WT ASXL2 or ASXL2 ΔMBH. Subunits of the MLL3/4 complex are highlighted in wheat. **c** A model showing that the ASXL2-MLL3/4 interaction bridges the MLL3/4 complex with the ASXL2-containing PR-DUB complex. **d** H2AK119 de-ubiquitination assay on a native chromatin substrate using WT ASXL2 and ASXL2 ΔMBH purified complexes shown in (**a**, **b**). (n = 1) Flag signals indicate similar amounts of WT and mutant complexes used, and H3 is the loading control. **e** H3K4 methylation assay using WT ASXL2 and ASXL2 ΔMBH purified complexes on recombinant human histone octamers or mono-nucleosomes. Flag signals show the amounts of complexes used in the reactions, and H4 is the loading control. (n = 2) (**f**, **g**) ChIP-qPCR analysis of H3K4me1 (**f**) and H2AK119ub (**g**) in K562 ASXL2 KD cells complemented or not with WT or ΔMBH ASXL2 stably expressed from the *AAVS1* locus. Two regions bordering an active enhancer upstream of the *GRHL2* gene were analyzed. Histone PTM levels were corrected for nucleosome occupancy (total H3 signal), presented as a ratio of IP/input (H3K4me1 or H2AK119ub/total H3). Data represent mean ± SEM from three biological replicates. Statistical analyses were performed by two-way ANOVA test followed by Tukey's test, $*p < 0.05$, $**p < 0.01$, $***p < 0.001$. Source data are provided in a Source Data file.

interaction between MLL3$_{PHD2/3}$ and ASXL2 may play a role in these malignancies.

Although structures and functions of MLL3$_{PHD2/3}$ and MLL4$_{PHD2/3}$ are conserved, we found differences in the regions adjacent to these modules. Analysis of amino acid sequences of MLL3 and MLL4 suggests that while MLL4$_{PHD2/3}$ is followed by a single zinc-knuckle, MLL3$_{PHD2/3}$ is followed by a zinc-knuckle and another PHD finger, the fourth one (MLL3$_{PHD4}$)[13]. AlphaFold modeling confirmed that MLL3$_{PHD4}$ (but not MLL4$_{PHD2/3}$) is rigidly linked to MLL3$_{PHD2/3}$ through a zinc-knuckle, and NMR titration experiments showed that MLL3$_{PHD4}$ does not make direct contacts with MLL3$_{PHD2/3}$ and does not interact with the ASXL2 peptide (Fig. 4f–i).

## MBH of ASXL2 directly links the native MLL3/4 and PR-DUB complexes in vivo

To determine the biological significance of the interaction between MLL3/4 and ASXL2 in cells, we engineered K562 cell lines expressing 3Flag-2Strep-tagged ASXL2 from the *AAVS1* safe harbor. Full length wild type ASXL2 and ΔMBH ASXL2 mutant lacking the MLL3/4-binding domain (aa 650-670) were studied in parallel. Tandem affinity purifications of wild type ASXL2 and ΔMBH ASXL2 from soluble nuclear extracts revealed that either wild type or mutated ASXL2 stably associates with the expected components of the PR-DUB complex, including the BAP1 subunit[29] (Suppl. Fig. 9). Western blot and mass spectrometry analyses of wild type ASXL2 further showed the co-fractionation of the PR-DUB complex with the components of the MLL3/4 complexes, including the H3K27me-specific demethylase KDM6A/UTX, as well as PAXIP1/PTIP, ASH2L, RBBP5, WDR5 and NCOA6 (Fig. 5a, b), supporting previous finding[16]. Strikingly, the removal of MBH in ASXL2 abrogated the interaction between ASXL2 and MLL3/4 in vivo (Fig. 5a, b). These data indicated that binding of MLL3/4$_{PHD2/3}$ to MBH of ASXL2 is solely responsible for the interaction between the two complexes, MLL3/4 and PR-DUB, in cells (Fig. 5c). These results also suggested that the interaction between MLL3/4 and ASXL2 may play a role in the BAP1-dependent protection of genomic regions from Polycomb-mediated silencing.

Since we were able to detect a native association of endogenous PR-DUB and MLL3/4 complexes in vivo and show the importance of the MBH of ASXL2 for this association, we investigated the impact of the deletion of MBH on enzymatic activities of these complexes. Using purified human chromatin as a substrate and equivalent amounts of purified fractions, we performed deubiquitinase assays followed by

MLL3. To validate this, we produced MLL3$_{PHD2/3}$ mutants reciprocal to the MLL4$_{PHD2/3}$ mutants described above and examined them by NMR and fluorescence spectroscopy (Fig. 4b–e and Suppl. Figs. 1, 6). Mutation of D372 or D400 to lysine in MLL3$_{PHD2/3}$ reduced binding to the ASXL2 peptide by ~40-fold and 400-fold, respectively, and the double mutation D372K/D400K completely eliminated this interaction in NMR titration experiments (Fig. 4b, e and Suppl. Fig. 1). The D372K/D400K MLL3$_{PHD2/3}$ mutant was also unable to bind full-length ASXL2 in GST pulldown assays (Suppl. Fig. 8). Similar to MLL4$_{PHD2/3}$, MLL3$_{PHD2/3}$ did not form the complex with the K661E ASXL2 peptide and showed reduced binding to R655I and I660E mutants of ASXL2 (Fig. 4b–d and Suppl. Fig. 6), confirming the conservation of the ASXL2-binding mechanism in MLL4$_{PHD2/3}$ and MLL3$_{PHD2/3}$.

Much like MLL4, MLL3 is commonly altered in cancer (Cosmic). D372 of MLL3 is found mutated to asparagine, tyrosine and valine in skin, prostate, large intestine, lung and upper aerodigestive tract cancers, E387 is mutated to lysine and valine in urinary tract cancer, and D400 is mutated to asparagine and histidine in large intestine, lung and upper aerodigestive tract cancers. Because the cancer-related D372N, D372K, E387Q and D400K mutants of MLL3$_{PHD2/3}$ showed reduced binding to ASXL2 (Fig. 4b), the disruption of the

immunoblot analysis. We detected efficient and equivalent de-ubiquitinition of H2AK119 by both wild type and ΔMBH ASXL2 purified complexes (Fig. 5d). This was also confirmed with fractions produced from transient co-transfection of ASXL2, wild type or ΔMBH, and BAP1 (Suppl. Fig. 10). We then measured the MLL3/4-specific activity in histone methyltransferase assays using recombinant histone octamers or nucleosomes as substrates. Mono-methylation of H3K4 was clearly detected with the purified wild type ASXL2 complex, whereas the H3K4me1-catalytic activity was markedly decreased in the case of the ΔMBH ASXL2 purified complex (Fig. 5e). These results confirmed the native MBH-dependent association of PR-DUB and MLL3/4 complexes in cells and also indicated that this binding does not alter each complex specific histone modifying activity.

To assess the effect of the MBH deletion in ASXL2 on epigenetic marks at an active enhancer we carried out chromatin immunoprecipitation (ChIP-qPCR) experiments (Fig. 5f, g and Suppl. Figs. 11, 12). A human K562 cell line, in which endogenous *ASXL2* expression was knocked down using CRISPR, was complemented with WT or ΔMBH ASXL2 expressed from the *AAVS1* safe harbor (Suppl. Fig. 11). H3K4me1, H2AK119ub, H3K27ac and H3K27me3 levels were measured on both ends of an active enhancer mapped upstream of *GRHL2* gene (ENCODE, Suppl. Fig. 12a). Expression of WT ASXL2 but not the ΔMBH ASXL2 mutant led to a decrease in H2AK119ub level at this enhancer, whereas the H3K4me1 level was not affected (Fig. 5f, g and Suppl. Fig. 12). These data revealed that MBH of ASXL2 is required for H2AK119 deubiquitination by BAP1 at this active enhancer in vivo but not required for H3K4 methylation by MLL3/4. These findings also suggest that MLL3/4 may recruit ASXL2/PR-DUB to protect the active state of the enhancer.

## MBH of ASXL1/2 is required for the association of BAP1 with MLL4 in ESCs

All homologous ASXLs (ASXL1, ASXL2 and ASXL3) associate with the catalytic BAP1 subunit of the PR-DUB complex via their DEUBAD domain (Fig. 1m). Following the in vitro identification and characterization of another binding partner of ASXLs, MLL3/4~PHD2/3~, we next explored whether the dual binding activity of ASXLs functionally links MLL3/4 and BAP1 in vivo. We used mouse embryonic stem cells (ESCs) because the sequences of human and mouse MLL3/4~PHD2/3~ and MBH of ASXL1/2/3 are highly conserved (Suppl. Fig. 13). Previous studies have shown that MLL4 knockout mice exhibit embryonic lethality at the embryonic day (E) 9.5, whereas MLL3 null mice die around birth[5]. Furthermore, the mRNA level of *Mll4* in ESCs is 3-fold higher than that of *Mll3* (Suppl. Fig. 14). These findings point to the predominant role of MLL4 over MLL3 in mouse embryogenesis and make ESCs a suitable system for evaluating the interplay between BAP1 and MLL4 in vivo.

As shown in Fig. 6a, *Asxl1* and *Asxl2* but not *Asxl3* are expressed in ESCs. To rule out the potential compensation involving ASXL1 and ASXL2, we designed guide RNAs (gRNAs) to delete the MBH region of ASXL1 and ASXL2. gRNAs, targeting *Asxl1* and *Asxl2*, were co-transfected into wild type (WT) ESCs for CRISPR/Cas9 gene editing (Fig. 6b). After screening ESC clones, two independent ESC lines were identified by genotyping and sequencing. These two ESC lines were characterized by homozygous deletion of or truncation prior MBH for ASXL1 and ASXL2 (hereafter referred to ΔMBH-1 and ΔMBH-2) (Fig. 6c, d). The MBH deletion did not affect ASXL1 or ASXL2 protein levels, and expression levels of other PR-DUB complex subunits, such as BAP1, FOXK1, HCF1, and OGT, as well as MLL4 were comparable among all cell lines (Fig. 6e). No noticeable changes in cell morphology were observed between WT and ΔMBH-1/2 cells (Fig. 6f). Following infection with lentiviruses expressing Doxycycline (Dox)-inducible T7-tagged BAP1, ESCs were treated with Dox for 1 day to induce robust and comparable BAP1-T7 expression (Suppl. Fig. 15). Consistent with the results in K562 cells (Fig. 5), disruption of the ASXL1/2 MBH regions did not compromise the integrity of the PR-DUB complex (Fig. 6g). In

WT cells, immunoprecipitation using an antibody against MLL4, followed by immunoblotting targeting T7 tag, showed that BAP1 was effectively co-immunoprecipitated with endogenous MLL4. However, co-immunoprecipitation of MLL4 and BAP1 was significantly reduced in ΔMBH-1 and ΔMBH-2 cells (Fig. 6h). These results indicated that BAP1 associates with MLL4 in ESCs in an ASXL1/2 MBH-dependent manner.

## MBH of ASXL1/2 links BAP1 to MLL4 on active enhancers in ESCs

To assess the genomic binding regions, we performed ChIP-seq analysis of BAP1-T7, MLL4, and histone modifications H3K4me1 and H3K27ac in ESCs expressing BAP1-T7. Among the 21,595 BAP1 binding regions identified in WT cells, 6286 (29%), 1842 (9%) and 3425 (16%) were located on active enhancers, primed enhancers, and promoters, respectively (Fig. 7a). Approximately 43% of BAP1+ promoters and ~65% of BAP1+ active enhancers were bound by MLL4, whereas less than 20% of BAP1+ primed enhancers or other regions were MLL4+ (Fig. 7b). These results revealed a substantial co-localization of BAP1 and MLL4 on promoters and active enhancers.

We next examined the recruitment of BAP1 and MLL4, as well as the distribution of H3K4me1 and H3K27ac, on BAP1+ MLL4+ promoters and active enhancers (Fig. 7c and Suppl. Fig. 16). Interestingly, BAP1 binding on active enhancers was markedly decreased in both ΔMBH-1 and ΔMBH-2 ESCs, while BAP1 binding on promoters was largely unaffected (Fig. 7c–e). Enhancer activation, indicated by the H3K27ac signal, was also compromised in the two ΔMBH cell lines (Fig. 7c–e). However, MLL4 recruitment and H3K4me1 intensity on active enhancers remained largely unchanged in ΔMBH-1 and ΔMBH-2 cells (Fig. 7c, e). Collectively, these data indicated that MBH of ASXL1/2 is required for BAP1 binding on active enhancers and suggested that MLL4 facilitates BAP1 binding on active enhancers via ASXL1/2 MBH.

In conclusion, MLL3 and MLL4 harbor eight and seven PHD fingers, respectively (Fig. 1a), which most likely act as scaffolding domains to enable these H3K4-specific methyltransferases to associate on demand with diverse binding partners depending on a specific chromatin state. While the majority of these PHD fingers remain structurally and functionally uncharacterized, in this study we demonstrate that MLL4~PHD2/3~ and MLL3~PHD2/3~ bind to the central α-helix of ASXL2 and that this binding plays a vital role in the direct interaction of the MLL3/4 complexes with the PR-DUB complex in vivo. Importantly, it is also required for the genomic co-occupancy of the deubiquitinase BAP1 and MLL4 on active enhancers, which in turn is essential for maintaining active state of these enhancers. We have previously shown that MLL4~PHD6~ and MLL3~PHD7~ recognize[14,15], a modification deposited by the acetyltransferase MOF that directs the recruitment of MLL4 to specific genomic sites. To better understand the multifaceted role of MLL3/4 in transcriptional regulation, future studies should be focused on establishing a mechanistic crosstalk between these three fundamental and functionally distinct chromatin-modifying complexes, MLL3/4, PR-DUB, and MOF.

Single PHD fingers and double PHD fingers (DPFs) are well-recognized readers of the histone H3 tail, either methylated (H3K4me3), acylated (H3K14acyl) or unmodified[30–35], and their binding activities have been shown to be crucial for various DNA-templated cellular processes. MLL4~PHD2/3~ and MLL3~PHD2/3~ represent a novel subset of PHD fingers that bind to non-histone proteins. Similarly, based on alignment of amino acid sequences, we foresee that other currently uncharacterized PHD fingers of MLL3 and MLL4 also have non-histone binding functions that are needed to be determined experimentally.

## Methods
### Protein expression and purification
The PHD2 and PHD3 fingers of human MLL3 (aa 342-439) and MLL4 (aa 227-324), the PHD4 finger of MLL3 (aa 464-520), and the linked MLL4~PHD2/3~-ASXL2 construct (aa 227-324 of MLL4, a SGPSSG linker, aa

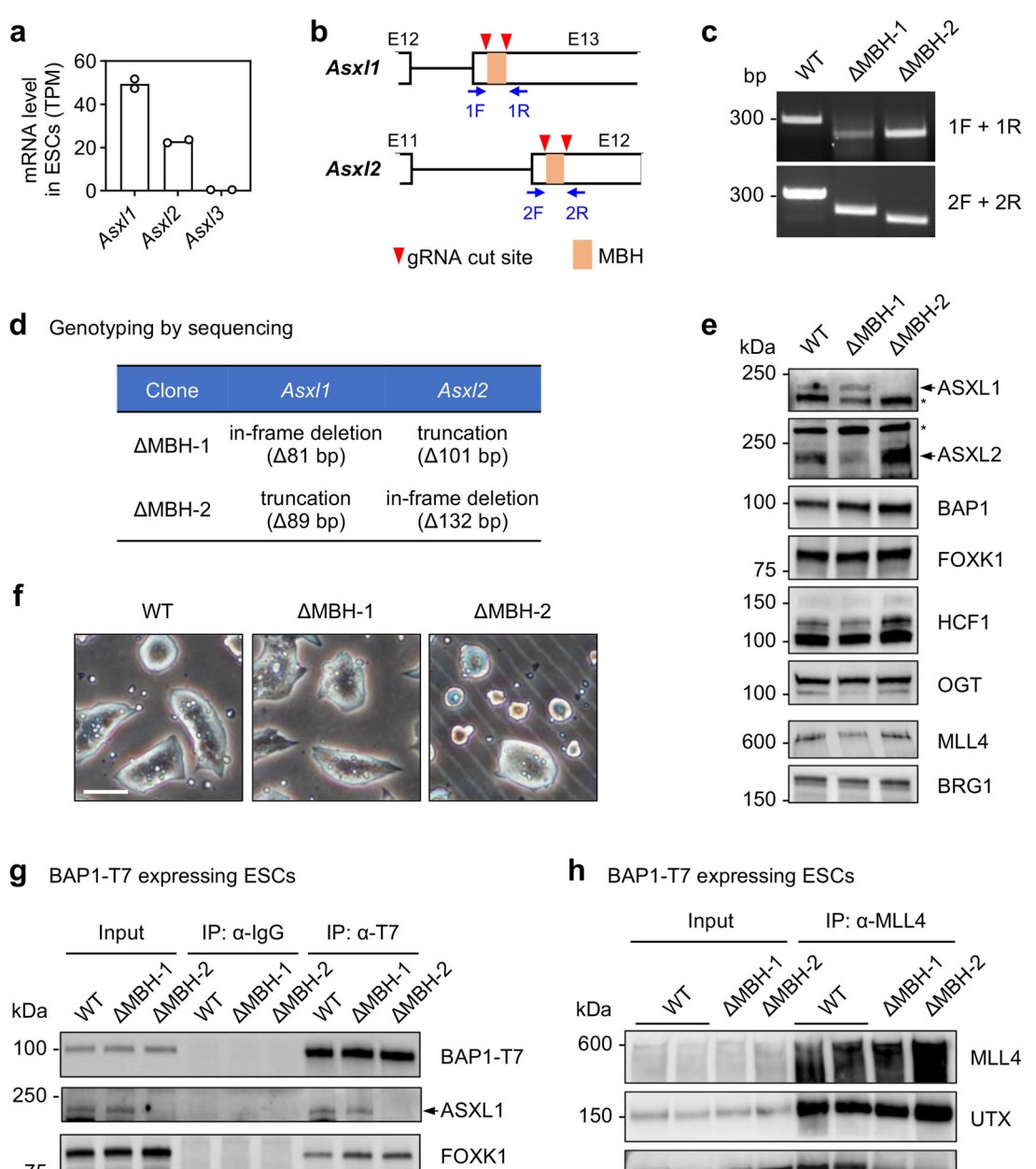

**Fig. 6 | MBH of ASXL1/2 is required for the association of MLL4 and BAP1 in ESCs. a** *Asxl1/2/3* mRNA levels in mouse ESCs. Data from RNA-seq (GSE154475)[1] are presented as dot plots (*n* = 2). Horizontal lines represent mean values. **b** Schematic representation of CRISPR/Cas9 gene editing-mediated deletion of ASXL1/2 MBH. Two cut sites flanking DNA sequences of MBH were induced by gRNAs inside exon 13 (E13) and exon 12 (E12) of *Asxl1* and *Asxl2*, respectively. Locations of PCR genotyping primers 1F, 1R, 2F, and 2R are indicated by arrows. **c** PCR genotyping using primer pairs indicated in (**b**). Sizes of PCR products are indicated on the left. **d** Summary of genotyping by sequencing results of two ASXL1/2 MBH deleted ESC clones, ΔMBH-1 and ΔMBH-2. **e** Whole cell lysates from wild type (WT), ΔMBH-1 and ΔMBH-2 ESCs were analyzed with immunoblotting using indicated antibodies. BRG1 is shown as a loading control. Asterisks indicate

non-specific bands. **f** Representative phase contrast microscopic images of WT, ΔMBH-1, and ΔMBH-2 ESCs. Scale bar, 50 μm. **g** WT, ΔMBH-1, and ΔMBH-2 ESCs were infected with Doxycycline (Dox)-inducible lentiviral vector expressing BAP1-T7. Cells were treated with 1 μg/ml Dox to induce BAP1-T7 expression. Whole-cell lysates were immunoprecipitated with IgG or anti-T7 antibody. Immunoprecipitates were analyzed by immunoblotting with antibodies indicated on the right. **h** The association of MLL4 and BAP1 is dependent on ASXL1/2 MBH. Whole-cell lysates from WT, ΔMBH-1, and ΔMBH-2 ESCs expressing BAP1-T7 were subjected to immunoprecipitation with anti-MLL4 antibody. Immunoprecipitates were analyzed by immunoblotting with antibodies indicated on the right. The experiments in (**c, e–h**) were performed independently at least twice. Source data are provided in a Source Data file.

650-670 of ASXL2) were cloned into a pGEX-6p-1 vector. All constructs were expressed in BL21-CodonPlus competent cells. Protein production was induced with 0.5 mM IPTG overnight at 16 °C in Luria broth (LB) or minimal media (M9) supplemented with $^{15}NH_4Cl$ and 0.05 mM

$ZnCl_2$ or $^{15}NH_4Cl$, $^{13}C$-glucose, and 0.05 mM $ZnCl_2$. The GST-tagged proteins were purified on Pierce™ Glutathione Agarose beads (Thermoscientific) in 50 mM Tris-HCl (pH 7.5) buffer, supplemented with 250 mM NaCl and 3 mM dithiothreitol (DTT). The GST tag was cleaved

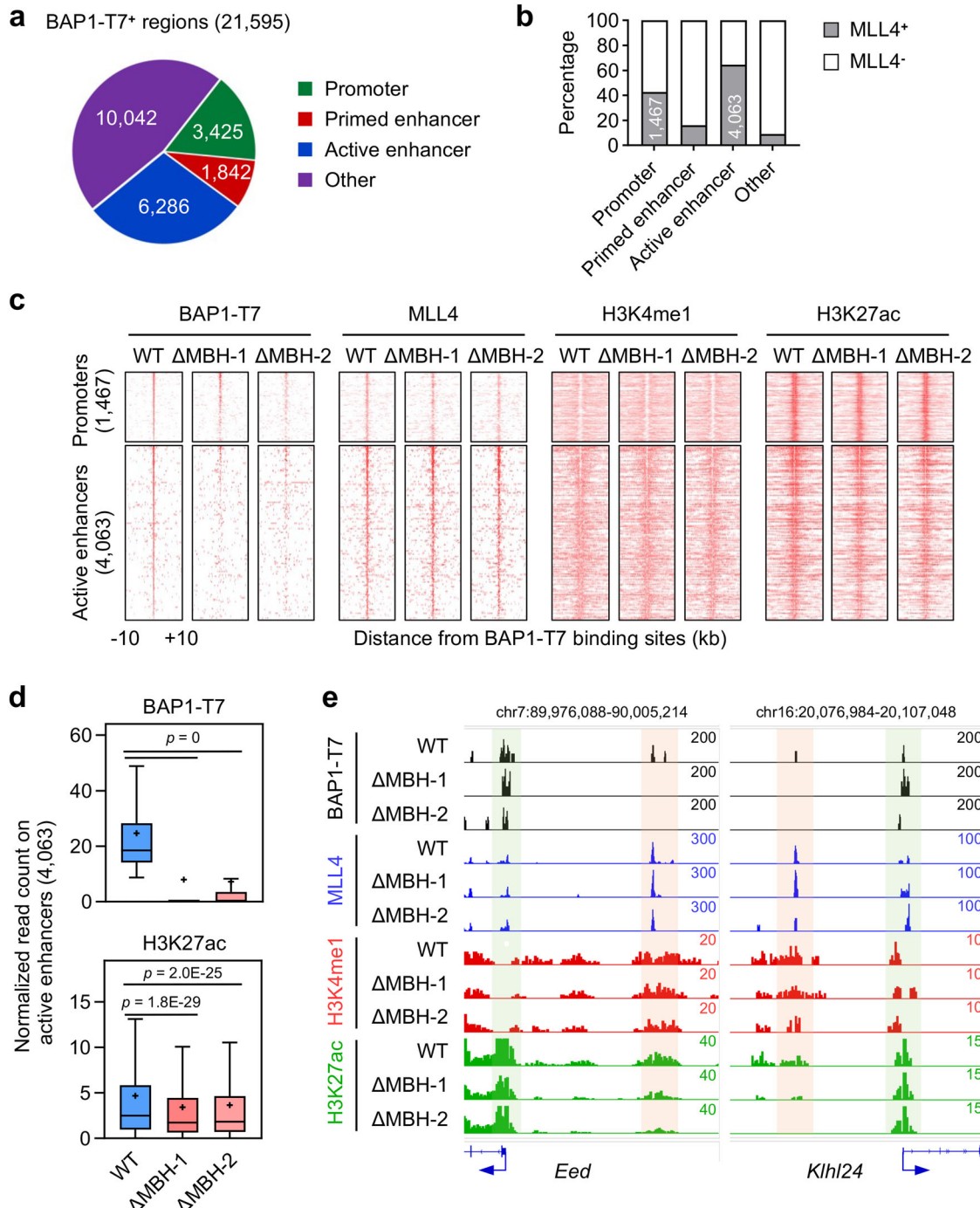

**Fig. 7 | MBH ASXL1/2 links BAP1 to MLL4 on active enhancers in ESCs. a** Genomic distribution of BAP1-T7 binding regions in WT ESCs expressing BAP1-T7. Promoters were defined as transcription start sites ±1 kb. Primed enhancers were defined as H3K4me1+ H3K27ac- promoter-distal regions. Active enhancers were defined as H3K4me1+ H3K27ac+ promoter-distal regions. The number of binding regions in each group is indicated. **b** Percentage of MLL4+ and MLL4- regions among BAP1-T7 binding regions as grouped in (**a**). Numbers of BAP1+ MLL4+ promoters and active enhancers are indicated. **c** Heat maps of BAP1-T7 and MLL4 genomic bindings as well as H3K4me1 and H3K27ac enrichments on BAP1+ MLL4+ promoters and active enhancers identified in (**b**). **d** BAP1-T7 and H3K27ac intensities on BAP1+ MLL4+ active enhancers (*n* = 4,063) in WT, ΔMBH-1, and ΔMBH-2 ESCs are presented in box plots. Center lines represent median values; crosses represent mean values; the bottom and top of the boxes represent lower and upper quartiles; whiskers were calculated using the Tukey method. Statistical significance was determined by the two-tailed Mann-Whitney *U* test. **e** ChIP-seq profiles of BAP1-T7, MLL4, H3K4me1 and H3K27ac in WT, ΔMBH-1, and ΔMBH-2 ESCs expressing BAP1-T7 are displayed on representative loci. BAP1-T7+ MLL4+ promoters and active enhancers are highlighted in light green and pink, respectively.

with PreScission protease overnight at 4 °C. Unlabeled proteins were further purified by size exclusion chromatography and concentrated in Amicon centrifugal filter units (MilliporeSigma). All mutants were generated by site-directed mutagenesis using the Q5 DNA polymerase and KLD Enzyme Mix (NEB), grown and purified as WT proteins.

**Phage display**

Phage display selections were performed in triplicate using the second-generation human disorderome M13 peptide-phage display library that displays 1 million 16 amino acid long peptides as described[36]. For each selection, bait protein (10 μg in 100 μL

phosphate-buffered saline (PBS, 137 mM NaCl, 2.7 mM KCl, 95 mM Na$_2$HPO$_4$, 15 mM KH$_2$PO$_4$ pH 7.5)) was immobilized in a well of a 96 well MaxiSorp plate (Nunc) for 18 h at 4 °C. For pre-selection depletion of non-specific binders, glutathione-S-transferase (GST, 10 μg/well) was immobilized in a separate plate. After immobilization, wells were blocked with 200 μL 0.5% (w/v) bovine serum albumin (BSA) in PBS for 1 h. Wells were washed four times with 200 μL PT buffer (PBS, 0.05% (v/v) Tween20). The phage library (10$^{11}$ pfu in 100 μL PBS) was incubated in a GST coated well for 1 h at 4 °C and then transferred to the bait coated well, and incubated for 2 h at 4 °C under gentle agitation. The phage solution was removed by aspiration and the well was washed 5 times with 200 μL PT buffer. Bound phages were eluted by incubation with 100 μL log-phase *E.coli* OmniMAX for 30 min at 37 °C, 200 rpm. The bacteria were hyper infected with M13KO7 helper phage (10$^9$ pfu/well, 45 min incubation) and then transferred to 1 mL 2YT (5 g NaCl, 16 g tryptone, 10 g yeast per liter) supplemented with 100 μg carbenicillin, 30 μg kanamycin and 0.3 mM IPTG. The bacteria were grown at 37 °C, 200 rpm for 16 h to allow for phage amplification. The bacteria were pelleted by centrifugation (3500 *g*, 10 min) and 800 μL of the phage supernatant was transferred to a new well and pH adjusted with 10 × PBS (90 μL). The phage solution was incubated at 65 °C to inactive any remaining bacteria before being used as in-phage library for the next round of selection. After 4 rounds of selection, the enrichment of binding phages was validated through phage pool ELISA. The peptide coding regions of binding enriched phage pools were amplified and barcoded, and sequenced using Illumina MiSeq v3, 1 × 150 bp read setup, 20% PhiX by the NGS-NGI SciLifeLab facility. NGS results processing was performed as described[36]. Peptide sequences were annotated using PepTools[37].

### NMR experiments
NMR experiments were carried out at 298 K on Varian INOVA 600 or 900 MHz spectrometers. NMR samples for structure determination using the linked MLL4$_{PHD2/3}$-ASXL2 (aa 227-324 of MLL4, a SGPSSG linker, aa 650-670 of ASXL2) construct were prepared in 20 mM Tris-HCl (pH 7.0) buffer, supplemented with 150 mM NaCl, 5 mM DTT and 7% D2O. Backbone and side chain chemical shift assignments for the MLL4$_{PHD2/3}$-ASXL2 complex were obtained by collecting and processing a set of triple resonance experiments (HNCACB, CBCA(CO)NH, CC(CO)NH, HBHA(CO)NH, HNCA, H(CCO)NH) with Non Linear Sampling on 2 mM $^{13}$C/$^{15}$N-labeled MLL4$_{PHD2/3}$-ASXL2, as in ref. 14. Chemical shift assignments were obtained for 96% of backbone amides. Three-dimensional $^{15}$N- and $^{13}$C-edited NOESY-HSQC spectra (mixing time of 100 ms) were collected to obtain distance restraints.

For NMR titration experiments, $^1$H,$^{15}$N HSQC spectra of 0.1 mM uniformly $^{15}$N-labeled WT or mutated MLL4$_{PHD2/3}$, MLL3$_{PHD2/3}$ or MLL3$_{PHD4}$ were recorded while WT or mutated ASXL2 (aa 650-670, 650-664, 656-670) and ASXL3 (aa 1047-1067) peptides (synthesized by SynPeptide) or MLL3$_{PHD2/3}$ were added stepwise. All experiments were carried out in 20 mM Tris buffer (pH 6.8), 150 mM NaCl, and 3 mM DTT, containing 7% D2O. The dissociation constants (K$_d$s) were determined by a nonlinear least-squares analysis in GraphPad Prism using the equation:

$$\triangle\delta = \triangle\delta_{max}\left(([L]+[P]+K_d) - \sqrt{([L]+[P]+K_d)^2 - 4[P][L]}\right)/2[P] \quad (1)$$

where [L] is concentration of the peptide, [P] is concentration of the protein, Δδ is the observed chemical shift change, and Δδ$_{max}$ is the normalized chemical shift change at saturation. Normalized chemical shift changes were calculated using the equation $\triangle\delta = \sqrt{(\triangle\delta H)^2 + (\triangle sn/5)^2}$ (2), where Δδ is the change in chemical shift in parts per million (ppm).

### Determination of the MLL4$_{PHD2/3}$-ASXL2 complex structure
Calculation of the structure of ASXL2-bound MLL4$_{PHD2/3}$ was carried out using interproton NOE-derived distance and dihedral angle restraints. NMR spectra were processed and analyzed with NMRPipe and CcpNmr. The program DANGLE in CcpNmr Suite was used to predict dihedral angle ψ and φ restraints. Hydrogen bonds were derived from characteristic NOE patterns in combination with dihedral angles. The zinc-coordinating residues were identified as described previously[14], and the structures were calculated and refined with XPLOR-NIH. Initially 200 structures were calculated using simulated annealing. 60 models with the lowest total energy were refined using XPLOR-NIH refinement scripts. An ensemble of 15 conformers with the lowest total energies were selected to represent the MLL4$_{PHD2/3}$-ASXL2 complex. The quality of the structures was validated using the program PROCHECK-NMR. The structural statistics are listed in Supplementary Table 1.

### Fluorescence spectroscopy
Spectra were recorded at 25 °C on a Fluoromax-3 spectrofluorometer (HORIBA), as described[38] with a few modifications. The samples containing 1.0 μM MLL4$_{PHD2/3}$ or MLL3$_{PHD2/3}$, WT or mutants, and progressively increasing concentrations of peptides were excited at 295 nm. Experiments were performed in buffer containing 20 mM Tris (pH 7.2), 150 mM NaCl, 1 mM TCEP and 0.02% Tween20. Emission spectra were recorded over a range of wavelengths between 330 and 360 nm with a 0.5 nm step size and a 0.5 s integration time and averaged over 3 scans. The K$_d$ values were determined using a nonlinear least-squares analysis and the equation:

$$\Delta I = \Delta I_{max} \frac{\left(([L]+[P]+K_d) - \sqrt{([L]+[P]+K_d)^2 - 4[P][L]}\right)}{2[P]} \quad (3)$$

where [L] is the concentration of the peptide, [P] is the concentration of protein, ΔI is the observed change of signal intensity, and ΔI$_{max}$ is the difference in signal intensity of the free and bound states. The K$_d$ value was averaged over three separate experiments, with error calculated as the standard deviation between the runs.

### GST pull-down assays
Pull-down assays were performed by mixing resin-bound GST-MLL3$_{PHD2/3}$ and GST-MLL4$_{PHD2/3}$ (wild type and mutants) with lysates of HEK293FT expressing Myc-ASXL1, Myc-ASXL2 or Myc-ASXL2ΔMBH. Pellets of HEK293FT cells transfected with the appropriate pDEST expression construct (15 μg/dish using polyethylenimine (PEI)) were lysed with 10 times their volume of lysis buffer (50 mM Tris pH 7.3, 100 mM NaCl, 1 mM DTT, 1 mM EDTA, 1 mM PMSF, 0.1% Tween20, 1/100 anti-protease (Sigma)) and sonicated with a probe sonicator (Branson digital sonifier 450) calibrated at 40% amplitude for 3–4 rounds during 40–60 s. Lysates were then centrifuged in a Sorvall Lynx 6000 (rotor: F21-8x50Y) at 15,000 RPM during 20 min. Approximatively 9 μg of purified GST-fusion proteins bound to GSH resin (Sigma, G4510) were used for each condition. Final volume of GSH resin was adjusted to 30 μl per condition with empty GSH beads and incubated with 1.5 mL of HEK293FT lysate expressing Myc-ASXL1, Myc-ASXL2 or Myc-ASXL2ΔMBH for 3 h at 4 °C on a rotative system. A fraction of the reaction mixture (80 μL) was collected as the input. After the incubation, beads were washed 5 times and mixed with 100 μL of Laemmli buffer. Samples were loaded on 8% or 15% SDS-acrylamide gels and processed for western blotting.

### Generation of tagged cell lines and purification of native PR-DUB complexes
Isogenic K562 cell lines expressing 3xFlag2xStrep-tagged ASXL2 WT and ΔMBH (aa 650-670), were generated by integration at the

AAVS1 safe harbor locus after DSB induction and recombination targeted by co-transfection with a ZFN expression plasmid, as previously described[39]. $2 \times 10^5$ cells were transfected with 400 ng of ZFN expression vector and 4 μg of donor constructs. Selection and cloning were performed in RPMI medium supplemented with 0.5 μg/mL puromycin starting 2–3 days post transfection. Clones were obtained by limiting dilution and expanded before harvest for western blot analysis.

For purification of native complexes, after large-scale expansion of K562 clones, affinity purifications of tagged ASXL2 WT and ΔMBH (aa 650-670) were performed on nuclear extracts as previously described[40]. Briefly, nuclear extracts were prepared following standard procedures and pre-cleared with CL6B Sepharose beads. FLAG immunoprecipitations with anti-FLAG agarose affinity gel (Sigma M2) were performed, followed by elution with 3xFLAG peptide (200 μg/mL from Sigma) in the following buffer: 20 mM HEPES pH 7.5, 150 mM KCl, 0.1 mM EDTA, 10% glycerol, 0.1% Tween20, 1 mM DTT and supplemented with proteases, deacetylases, and phosphatase inhibitors. Eluted fraction was then mixed with Strep-Tactin Sepharose (IBA) affinity matrix for 2 h at 4 °C, and, after washes, complexes were eluted with same buffer supplemented with 4 mM D-biotin, flash-frozen in liquid nitrogen, and stored at −80 °C. Typically, 5–15 μl of the elutions (1–3% of total) was loaded on NuPAGE 4–12% Bis-Tris gels (Invitrogen) and analyzed by silver staining and western blots. Antibodies used include: MLL3 (Cell Signaling 53641), BAP1 ((C-4) Santa Cruz sc-28383), UTX (Cell Signaling, 33510), FLAG (Sigma, A8592). Purified fractions were prepared and analyzed at the Proteomics Platform of the Quebec Genomic Center using an Orbitrap Fusion mass spectrometer (Thermo Fisher Scientific) as previously described[41]. Raw data were deposited at the MassIVE public repository under the accession numbers MSV000093435 and PXD047103.

## Deubiquitination assays using native PR-DUB complexes

The deubiquitination reactions were done by adding equivalent amounts of ASXL2 WT or ΔMBH (4 μl E1-F) native complexes purified from K562 cells to 0.5 μg of chromatin enriched extracts from K562 cells, incubated at 37 °C in 50 mM Tris-HCl pH 7.3, 50 mM NaCl, 1 mM $MgCl_2$, 1 mM DTT, 1 mM PMSF, 10 μg/ml Leupeptin, 10 μg/ml Pepstatin A, 25 μg/ml Aprotinin. After 3 or 6 h incubation, 4X Laemmli buffer was added, the samples run on 15% SDS-PAGE and transferred on nitrocellulose membrane. Immunoblotting was performed with an anti-H2AK119ub (Cell Signaling, D27C4, rabbit polyclonal) diluted 1:2000 followed by HRP conjugated secondary goat anti-rabbit antibody (Sigma) used at a 1:10,000 dilution. The immunoblots were visualized using a Western Lightning plus-ECL reagent (Perkin-Elmer). Anti-H3 (Abcam, Ab1791, rabbit polyclonal) was used at a 1:50,000 dilution.

## In vitro histone methyltransferase (HMT) assays

Equivalent amounts of purified ASXL2 WT and ΔMBH complexes were incubated with 0.5 μg of reconstituted recombinant mononucleosomes or histone octamers as substrates[42]. All HMT assays were performed in 50 mM Tris-HCl pH 8, 100 mM KCl, 5 mM $MgCl_2$, 0.1 mM DTT, 10% glycerol in the presence of 50 μM S-adenosyl-L-methionine (Sigma, A7007). After an 8 h incubation at 30 °C, 4X Laemmli buffer was added, the samples run on 15% SDS-PAGE and transferred on nitrocellulose membrane. Immunoblotting was performed with an anti-H3K4me1 (Ab8895, rabbit polyclonal) diluted 1:1000 followed by HRP conjugated secondary goat anti-rabbit antibody (Sigma) used at a 1:10,000 dilution. The immunoblots were visualized using a Western Lightning plus-ECL reagent (Perkin-Elmer). Anti-H4 (Abcam, Ab7311, rabbit polyclonal) was used at a 1:2,500 dilution.

## BAP1/ASXL2 purification and DUB assay

For each BAP1/ASXL2 complex, 6 dishes of HEK293FT cells were transfected with 3 μg of pDEST-HA-BAP1 or pDEST-HA-BAP1 C91S and

30 μg of pDEST-Flag-ASXL2 or pDEST-Flag-ASXL2ΔMBH. Cell pellets were harvested 2 days post-transfection and lysed with 10 times their own volume with EB300 lysis buffer (50 mM Tris pH 7.3, 300 mM NaCl, 0.5% Triton X100 (V/V), 5 mM EDTA, 50 mM NaF, 1 mM $Na_3VO_4$ (activated), 10 mM Beta-glycerol-phosphate, 1 mM DTT, 1 mM PMSF, 1/100 anti-protease (Sigma)). Lysates were incubated on ice for 30 min before centrifugation in a Sorvall Lynx 6000 (rotor: F21-8x50Y) at 15,000 RPM during 20 min. Supernatants were collected and diluted with EB0 (50 mM Tris pH 7.3, 0.5% Triton X100 (V/V), 5 mM EDTA, 50 mM NaF, 1 mM $Na_3VO_4$, 10 mM Beta-glycerol-phosphate, 1 mM DTT, 1 mM PMSF, 1/100 anti-protease) to adjust the final concentration of NaCl to 150 mM. Agarose anti-FlagM2 beads (40 μL) (Sigma, A2220) were added to each lysate and incubated overnight at 4 °C on a rotative system. The next day, beads were washed 5 times with the wash buffer (50 mM Tris pH 7.3, 150 mM NaCl, 0.5% Triton X100 (V/V), 1/500 anti-protease, 1 mM DTT, 1 mM PMSF). The BAP1/ASXL2 complexes were then eluted from the anti-FlagM2 beads using 200 μL of Flag peptide elution buffer (50 mM Tris pH 7.3, 150 mM NaCl, 1/500 anti-protease (sigma), 1 mM DTT, 1 mM PMSF, 0.2 mg/mL Flag peptide). Following three elutions of 2 h each, the pooled fractions were incubated with 40 μL of anti-HA beads (Sigma A2095) overnight at 4 °C on a rotative system. The next day, the beads were washed 5 times with the wash buffer, and the protein complexes were then eluted 3 times with 200 μL of HA peptide elution buffer (50 mM Tris pH 7.3, 50 mM NaCl, 1/500 anti-protease (Sigma), 1 mM DTT, 1 mM PMSF, 0.2 mg/mL HA peptide). Next, the HA peptide elution buffer was replaced with a conservation buffer (50 mM Tris pH 7,3, 50 mM NaCl, 1 mM DTT, 1 mM $MgCl_2$, 10% Glycerol) using a 30,000 Da centricon unit (Millipore). Aliquots of the purifications were then flash frozen in dry ice and stored at −80 °C.

The deubiquitination assays on native nucleosomes were done using the BAP1/ASXL2 complexes purified from HEK293FT cells. These complexes were incubated with 3 μg of nucleosomes purified from HEK293FT cells. The reaction was performed at room temperature with the final volume set at 100 μL. DUB buffer was composed of 50 mM Tris pH 7.3, 50 mM NaCl, 1 mM $MgCl_2$, 1 mM DTT. The DUB reaction was stopped at the indicated time points by mixing 30 μL of the reaction mixture with 30 μL of Laemmli buffer and processed for western blotting.

## K562 cell lines for ChIP-qPCR

K562 cell lines with knocked down expression of endogenous ASXL2 were obtained after double infection of lentivirus expressing Cas9 and 4 different gRNAs targeting the end of exon 1 (pLENTI_CRISPR_V2, co-selection by FACS with GFP). Clone 1 L was selected (obtained with gRNA 1: 5′-CTCCCTCCCCCTTACCGTCT-3′ cutting at the very end of exon 1) for transfection of WT and DMBH ASXL2 vectors for integration and stable expression from the *AAVS1* safe harbor, as described for the tagged K562 cell lines used for purification. Expression levels of endogenous ASXL2, WT and ΔMBH ASXL2 were monitored by running SDS-PAGE and transferring onto nitrocellulose membrane. Anti-FLAG M2 conjugated to horseradish peroxidase (A8592, Sigma) was used at 1:10,000 dilution and anti-ASXL2, at 1:2,000 (Cell Signaling (E623X), CS71257, lot:1). Immunoblots were visualized using a Western Lightning plus ECL reagent (Perkin-Elmer).

## ChIP-qPCR for K562 cell lines

Crosslinked chromatin from K562 cells and immunoprecipitations were prepared as previously described[43]. For chromatin immunoprecipitation, 100 μg of chromatin was incubated with 2 μg anti-H2AK119ub (Cell Signaling, 8240S, D27C4 lot:9) or 3 μg anti-H3 (Abcam, ab1791, lot GR3421644-1), 2 μg anti-H3Kme1 (Abcam, ab8895, lot GR442591-1), 2 μg anti-H3K27ac (Thermo Fisher, MA5-23516, lot: ZA4181671) or 0.3 μg anti-H3K27me3 (Cell Signaling, C36B11, 9733 S lot:19) antibodies overnight at 4 °C. 25 μl of Protein A Dynabeads were then added to each sample, and the mixtures were

incubated at 4 °C for 4 h. The beads were washed extensively and eluted with 1% SDS and 0.1 M NaHCO3. Cross-linked samples were reversed by heating overnight at 65 °C in the presence of 0.2 M NaCl. Samples were then treated with RNase A and proteinase K for 2 h, and DNA was recovered using MinElute PCR purification Kit (Qiagen, 28004) according to the manufacturer's instructions. Quantitative real-time PCR corrected for primer efficiencies in the linear range was performed using SYBR Green I (Roche, 04877352001) on a LightCycler 480 (Roche). Primers for 5′ end of active enhancer: Fwd: 5′-ACCC TAACAATCTCTGCCGAC; Rev: 5′-TTGAGAACAGCATGTGGGGT; For 3′ end: Fwd: 5′-CTCAGGAATTTGGGCGGAGT, Rev: 5′-GCGTCACACTTG ACGATCCA.

## Generation of ΔMBH mouse ESCs

Mouse ESCs were cultured as described previously[1]. ΔMBH ESC lines were generated using the CRISPR/Cas9 system. Briefly, gRNA sequences flanking ASXL1 MBH (5′-ATCACGGAGTCCTCCTGCCG-3′ and 5′-TGGCCGTCTCTCGATTGCAG-3′) or flanking ASXL2 MBH (5′-ATCACTAGTCCTTACAGAAC-3′ and 5′-AATGGTCCCTCCAACTGA GG-3′) was cloned into the lentiCRISPR v2 plasmid (Addgene #52961). Plasmids containing 4 gRNAs were co-transfected into wild type V6.5 ESCs by Lipofectamine 3000 (Thermo Fisher). 24 h later, cells were selected with 1.5 µg/mL puromycin for 3 days. After recovery, $1 \times 10^4$ single cells were seeded into a 6 cm dish. 1 week later, colonies from single cells were picked up and expanded for genotyping. ESCs harboring MBH deletion for both ASXL1 and ASXL2 were verified by Sanger sequencing.

## Generation of cell lines expressing BAP1-T7

Wild type or ΔMBH ESC clones were infected with Doxycycline-inducible lentiviral pCW57.1 vector (Addgene #41393) expressing triple T7-tagged BAP1 (BAP1-T7). 2 days later, cells were selected with 1.5 µg/ml puromycin for 3 days. After recovery, cells were treated with 1 µg/mL Doxycycline for 24 h and collected for immunoblotting, immunoprecipitation, and ChIP-seq.

## Immunoblotting and immunoprecipitation

To extract whole cell lysates, cells were collected and lysed in RIPA buffer (50 mM Tris-HCl pH 8.0, 150 mM NaCl, 0.1% SDS, 1% NP-40, 0.5% sodium deoxycholate) supplemented with protease inhibitors and 0.25 U/mL Benzonase nuclease for 40 min. Whole cell lysates were resolved using 4–15% TGX™ precast protein gels (Bio-Rad).

Co-immunoprecipitation (IP) was performed as described previously[44]. Briefly, cells were lysed in IP buffer (50 mM Tris-HCl pH 7.5, 150 mM NaCl, 0.3% NP-40, and 2 mM EDTA) supplemented with protease inhibitors for 30 min at 4 °C, and then incubated overnight with Dynabeads Protein A (Thermo Fisher) pre-bound by anti-MLL4 antibody. Beads were subsequently washed 5 times using IP buffer. Immunoprecipitated complexes were eluted by boiling the beads for 10 min and resolved using NuPAGE™ 3–8% Tris-Acetate gels (Invitrogen), as described[1].

## Antibodies

Anti-BRG1 (ab110641), anti-FOXK1 (ab18196), anti-OCT4 (ab19857) and anti-H3K27ac (ab4729) were from Abcam. Anti-T7 (D9E1X, 13246), Anti-H2AK119ub (D27C4), Anti H3 (1B1B2) and Anti-ASXL2 (E6Z3X) were from Cell Signaling Technology. Anti-ASXL2 (A302-037A), anti-HCF1 (A301-400A), and anti-RbBP5 (A300-109A) was from Bethyl Laboratories. Anti-ASXL1 (GTX127284) was from GeneTex. Anti-OGT (sc-32921) was from Santa Cruz. Anti-H3K4me1 (13-0040) was from EpiCypher. Anti-UTX, Anti-HA, Anti-Myc, and anti-MLL4 antibodies were homemade.

## ChIP and ChIP-seq library preparation

Cells were cross-linked with 1% formaldehyde for 10 min and quenched by 125 mM glycine for 10 min. Fixed cells were swelled in the lysis buffer (5 mM PIPES pH 7.5, 85 mM KCl, 0.5% NP-40) supplemented with protease inhibitors for 20 min, and centrifuged at $500 \times g$ for 5 min at 4 °C. Nuclei were resuspended with cold TE buffer (10 mM Tris-HCl pH8.0, 1 mM EDTA), and subjected to sonication. Sheared chromatin was clarified by centrifugation at $13,000 \times g$ for 10 min at 4 °C. The supernatant was transferred to a new tube and further complemented to 150 mM NaCl, 1% Triton X-100, 0.1% sodium deoxycholate and protease inhibitors. 2% of the mixture was set aside as input. For each ChIP, chromatin from $1 \times 10^7$ cells were mixed with 20 ng spike-in chromatin (Active Motif, #53083) and incubated overnight at 4 °C with Dynabeads Protein A (Thermo Fisher) pre-bound by 2–8 µg primary target antibody and 2 µg spike-in antibody (Active Motif, #61686). Beads were washed twice with 1 mL RIPA buffer, once with 1 mL RIPA buffer containing 500 mM NaCl, twice with 1 mL LiCl buffer, and once with 1 mL TE buffer, and then eluted with 100 µL fresh elution buffer (0.1 M NaHCO₃, 1% SDS). ChIP samples and input were incubated with Proteinase K (NEB) at 65 °C overnight to reverse formaldehyde cross-linking, and then purified using QIAquick PCR purification kit (Qiagen).

For ChIP-Seq, the entire ChIP DNA and 300 ng input DNA were used to construct libraries using NEBNext Ultra™ II DNA Library Prep kit for Illumina (NEB, E7645) following the manufacturer's instructions. The final libraries were sequenced on Illumina NovaSeq 6000.

## Computational analysis

**NGS data processing.** Raw sequencing data were aligned to the mouse genome mm10 and the drosophila genome dm6 using Bowtie2 (v2.3.4)[45]. To identify ChIP-enriched regions, SICER (v2) was used (https://zanglab.github.io/SICER2/). For ChIP-Seq of T7 tag and MLL4, the window size of 50 bp, the gap size of 50 bp, and the false discovery rate (FDR) threshold of $10^{-10}$ were used. For ChIP-Seq of histone modifications H3K4me1 and H3K27ac, the window size of 200 bp, the gap size of 200 bp, and the FDR threshold of $10^{-3}$ were used. Reads on indicated regions were collected to calculate Reads Per Kilobase Million (RPKM) as a measure of signal intensity.

**Analysis of ChIP-seq peaks.** To define regulatory regions, a combination of genomic coordinates and histone modification ChIP-seq data were used. Promoter regions were defined as transcription start sites ± 1 kb. Promoter-distal regions were further overlapped with H3K27ac⁻ regions or H3K27ac⁺ regions to define primed enhancers and active enhancers, respectively. Target⁺ regions were defined as regions that are overlapping with target peaks for at least 1 bp.

Heat map matrices were generated using in-house scripts with 50 bp resolution and visualized in R (v4.3.1). Genomic regions shown in heat maps were ranked according to the intensity of BAP1-T7 at the center of the 400 bp window in WT cells.

**Genomic profile visualization.** To visualize data from ChIP-seq in the Integrative Genomics Viewer (IGV) (v2.16.2)[46], wiggle (wig) format-profiles generated from SICER (v2) were used.

## Reporting summary

Further information on research design is available in the Nature Portfolio Reporting Summary linked to this article.

## Data availability

Coordinates for the MLL4_{PHD2/3}-ASXL2 complex have been deposited in the Protein Data Bank under accession number 9ATN. NMR parameters for the MLL4_{PHD2/3}-ASXL2 complex have been deposited to the Biological Magnetic Resonance Data Bank under accession number 31041. Raw data were deposited at the MassIVE and ProteomeXchange public repository under the accession number MSV000093435 and PXD047103, respectively. The password to access the MS files prior to publication is "PHD" at ftp://MSV000093435@massive.ucsd.edu. All data sets described in the paper have been deposited in NCBI Gene

Expression Omnibus under accession number GSE248027. The mm10 was used as the mouse genome reference. All other relevant data supporting the key findings of this study are available within the article, its Supplementary Information or from the corresponding authors upon reasonable request. Source data are provided in the Source Data file. A reporting summary for this article is available as a Supplementary Information file. Source data are provided with this paper.

## Code availability

This paper does not report original code. No custom code package or newly developed algorithm were generated in this study. All code used for the analysis are available from the corresponding author upon reasonable request.

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

## Acknowledgements

We thank Céline Roques, Yannick Doyon, and Lara Cardinaels for technical support, Maxime Galloy and Amélie Fradet-Turcotte for reagents, and NHLBI DNA Sequencing and Genomics Core for next generation sequencing. This work was supported by grants from NIH GM125195, GM135671, HL151334, CA252707, and AG067664 to T.G.K., the Intramural Research Program of NIH NIDDK to K.G., NIH K99CA241301 to Y.Z., the Canadian Institutes of Health Research (CIHR) (FDN-143314) to J.C. and CIHR MOP399244 to EBA, and the Institute of Biomedical Sciences, Academia Sinica, and the Ministry of Science and Technology in Taiwan to S.P.W. This study made use of NMRbox: National Center for Biomolecular NMR Data Processing and Analysis, a Biomedical Technology Research Resource (BTRR), which is supported by NIH grant P41GM111135.

## Author contributions

Y.Z., G.X., J.E.L., M.Z., D.S., B.E., C.B., T.V., G.A., C.L., C.M., L.S., V.K.S., J.P.L., and Y.W.C. performed experiments and together with S.P.W., Y.I., E.B.A, J.C., K.G., and T.G.K. analyzed the data. G.X., J.E.L., K.G., and T.G.K. wrote the manuscript with input from all authors.

## Competing interests

The authors declare no competing interests.
