## [Peer Review File · Nature Communications]

ASXLs binding to the PHD2/3 fingers of MLL4 provides a mechanism for the recruitment of BAP1 to active enhancersReviewers' comments:

Reviewer #1 (Remarks to the Author):

In the manuscript entitled "ASXL2 links MLL4 and BAP1 on enhancers", the authors reported the identification of an MLL-binding helix (MBH) domain within ASXL2 protein, which links MLL3/4 to the BAP1 complex. They then further identified a subset of genes that were co-occupied and co-regulated by MLL4 and BAP1.

Major concerns-

In Line 130, "conserved function of MLL3PHD2/3 and MLL4PHD2/3 that can bridge the H3K4-specific methyltransferase MLL3/4 complexes with the H2AK119ub-specific deubiquitinase PR-DUB complex." The authors need to deplete the HBM motif from the full-length ASXLs, and further evaluate the protein-protein interaction between endogenous MLL3/4 (not just the PHD domains) and wild-type/HBM-depleted ASXLs.

In Figure 3F, the authors need to explain why they used leukemia cell line HL60 for the IP experiment between endogenous ASXL2 and MLL3/4 PHD domains? In addition, they need to provide protein levels for input samples to show that equal levels of wild-type/mutant PHDs were expressed in cells. However, in Figure 3K-M, the authors switched to HEK293T cells to study the chromatin occupancy of flag-tagged mutant ASXL1, which contains leukemia-specific truncated mutations. In addition, the MLL3/4 ChIP-seq should be included to determine the impact of mutant ASXL1 on MLL3/4 recruitment.

In Figure 3I, the results from the clinical studies were not convincing. The p-value needs to be provided for each of the analysis.

In Fig 4K, the peptide number of MLL3 was 10 times higher than MLL4, suggesting that MLL3/ASXL2 interaction is much stronger than MLL4/ASXL2, which is not in agreement with their NMR results. Also, the subunits RBBP5 and DPY30 were missed in the mass spec results, which are critical for the catalytic activity of MLL3/4. The authors need to conduct the in vitro methylation assay to show that the purified MLL3/4 complex by ASXL2 still has lysine methyltransferase activity.

In Figure 5E, based on the Venn-diagram analysis, only 4.5% (2,843 of 62,590) of BAP1 peaks were co-localized with MLL4. The authors need to conduct a ChIP-seq experiment using T7 antibody in wild-type cells as a negative control (Input or IgG is not be a good control here). In addition, the authors also need to conduct ASXL2 ChIP-seq and compare MLL4 peaks with BAP1/ASXL2 overlapping peaks.

In Figure 5F, it seems that BAP1's occupancy is also reduced at BAP1+/MLL4+ common loci in MLL4-KO cells. It is necessary to include H2AK119Ub levels here.

My major concern comes from the cell line the authors used for their genome-wide studies (Figure 5-7). It has already been known that there is no detectable MLL3 protein in HCT116 cells. However, at least one copy of MLL4 gene is also mutated (P2442fs and V160M, based on COSMIC and DepMap Data) in this cell line, but it is still not clear whether these mutations are gain-of-function or loss-of-function mutations. Therefore, HCT116 cell line is not an appropriate model to establish for their NGS studies.

Other suggestions-

In Fig 4J, ASXL2 protein levels should be included.

Reviewer #2 (Remarks to the Author):

In this comprehensive study, Zhang et al. identified a critical interaction between MLL3/4 and ASXL protein family and demonstrated that such interaction plays an important role in co-localizing MLL3/4 complexes and PR-DUB complex on promoters and enhancers to regulate gene expression. The authors first applied a combination of techniques including phage display, NMR, and fluorescence spectroscopy to discover and characterize a binding interface mediated by MLL3/4 PHD2/3 and ASXL-MBH, and then identified a specific set of genes that are likely regulated by such interaction via ChIP-Seq. The evidences provided are solid and convincing, and overall, the study is of great interest to the field and provides insights into the regulation of MLL3/4 and PR-DUB complexes in human diseases such as cancer.

I have the following minor concerns that should be addressed to improve the manuscript.

1. The authors screened a peptide library which contains 1 million peptides for binding MLL3 PHD2/3 using phage display and two hits were identified (Fig 1C). Are they the only hits? If not, are there other potential proteins that may also interact with MLL3 PHD2/3? It seems like Fig 1C was not explained well in the text or legend.
2. For NMR-HSQC data, (i) since the residues of MLL3 (or 4) PHD2/3 are assigned, I encourage the authors to provide a chemical shift change (or contour level change) profile for each residue of MLL3 (or 4) PHD2/3 upon ASXL binding. That helps visualize the key residues involved in the binding interface. (ii) For selected HSQC spectra, perhaps arrows could be applied to indicate the shift of the peaks and some key peaks could be labeled with residues if assigned.
3. I suggest supplementary FigS2 be moved to main Figure 2. Zinc fingers residues and interface residues can be highlighted since they are mentioned throughout the manuscript.
4. MLL3/4 contains several PHD domains, are they similar? Have the authors tested their binding to ASXL peptides? A sequence alignment of the PHD domains especially the interface residues could help elucidate the specificity of PHD2-3/ASXL interaction.
5. For supplementary FigS3, it seems like the contour levels for each spectrum are not normalized. Line 138-140 "The 1H,15N HSQC spectrum of the linked MLL4PHD2/3-ASXL2 construct overlaid well with the spectrum of the ASXL2-bound unlinked MLL4PHD2/3 construct" should be rephrased since they are not exactly overlapped. Key residues could be labeled on the spectra.
6. The authors determined the solution structure by NMR but there is little description regarding the starting template, number of NOEs assigned for structural calculation, and percentage of residue assignments in the text. I also recommend the authors to provide an overlap of structures with top 10 or 15 energy states calculated. Based on the supplementary table 1, the backbone and all heavy atoms RMSDs are estimated to be 1.6 Ang and 1.1 Ang, which are a bit high. Did they include unstructured regions of proteins in the RMSD estimations?
7. For structural illustration, e.g. Figure 2d and 3a, (i) H-bonds could be labeled using dashed lines to help readers visualize the interactions. (ii) Zinc finger residues could also be displayed since they are mentioned in the text. (iii) It's hard to see how E272K mutation affect the binding based on Fig 3a. (iv) Line 159-160, Residues "652-654" are not labeled in the figure. What kind of interactions does this portion of ASXL2 mediate?
8. For figure 3f, the input of GST-tagged MLL3 variants should be provided. Can the authors explain why ASXL2 in pull-down group shows two bands but input group only shows one?
9. Line 191-194, did the authors test if the C276Y and C294R of MLL3 or G653E of ASXL2 impair the binding? If not, they should rephrase the text (Line 196-198) "Overall, these findings suggest that the impaired interaction between MLL4PHD2/3 and ASXL2 could contribute to the development of solid cancers."
10. Does deletion of MBH affect the overall structure/function of ASXL2 besides MLL3 binding?
11. Based on the Chip-seq data, some genes are regulated by MLL3 and BAP1 but independent of

AXSL2?

12. Both MLL3/4 and AXL2 are multi-domain containing proteins that form large complexes with a variety of proteins. It would be helpful for readers to better/quickly understand if the authors could label the residue numbers for all domain in the domain-diagram (Figure 1A and 1M) and also indicate the identified key binding partners in the figure.

Reviewer #3 (Remarks to the Author):

Based on careful biochemical characterizations of PHD2/3 fingers of MLL3 and MLL4, this manuscript by Zhang et al. links the COMPASS complex to PR-DUB. In addition to confirming the physical interaction between ASXL proteins with MLL3/4, the authors have demonstrated that the two complexes are interdependent at co-localized enhancers and may co-regulate the transcription of a subset of target genes. This study has provided a clear structural basis for the interaction between the two complexes. But quite a few concerns discourages us from recommendation of its publication in the current form.

Major comments:

1. There lacks genomic and functional analyses of MBH. The identification of MBH is the most interesting finding in this study. However, the authors did not center this domain to design experiments. First, MBH's necessity seems different in solid cancers and hematopoietic malignancies. But the authors did not even discuss about the underlying cause for these differences. Second, the only experimental data showing the importance of MBH are shown in Figure 3k-m. These data are generally weak and not that convincing. Besides, the data show that ASXL1 is localized at TSS in a MBH-dependent manner, which is inconsistent with the dominant colocalization of COMPASS complex and PR-DUB at enhancers. Third, no rescue experiments in ASXL2KO cells with either WT or MBH deletion/mutation were designed. The ChIP-seq analyses even functional analyses with these rescues are critical to claim the importance of MBH.
2. We are highly concerned about the BAP1-T7/MLL4 ChIP-seq data, as shown in Figures 5-7. The TPM/FPKM values and genomic localizations should be labelled on all the tracks. The comparison should be group auto-scaled. However seeing from the tracks and accessible bigwig files, the BAP1-T7/MLL4 peaks do not look right. The sharp peaks without any genomic background are very likely due to a failed library preparation. In this condition, the difference between control and KO groups could be random. The authors should do a careful QC for the original data. And MLL4 ChIP-seq in MLL4KO cells to confirm the signal specificity would be helpful.
3. In addition to the uncertain data quality, the data analyses were poorly performed. Here are a few examples. 1) As we know, ASXL1/2 mutations are frequently found in various subtypes of myeloid malignancy with totally different prognosis. However in Figure 3i, the authors did not distinguish the subtypes and failed to observe the significant difference in prognosis for ASXL1 truncation mutants with MBH or not. The authors are encouraged to make a careful analysis within each subtype of malignancies. 2) In Figures 5 and 6, the authors only show the positive data supporting the interdependency at MLL4/BAP1 co-occupied regions, without convincing negative controls (BAP1 or MLL4-negative regions). Actually looking from the tracks, MLL4 signals are also decreased or lost from BAP1-negative regions. It is very likely that these changes are just random, or secondary to cell fate changes or due to alternative mechanisms. To have a clear conclusion in here, optimal data quality and the rescue experiments as mentioned above are required.
4. MLL3/4 and PR-DUB complexes are well known for their biochemical activities on histones. To establish the functional links, the authors should show the effects on H3K4me1 and H2AK119Ub1 at the genome wide in either MLL4 or PR-DUB deficient cells. Without these data, we can hardly understand the biological significance for the two coupled complexes in enhancer regulation.
5. Are there any effects on cell growth after deletion of BAP1 or MLL4 in HCT116 cells? BAP1 deletion usually leads to cell growth defects, which possibly makes the subclones different. Looking from Figure 5d, H3K4me1 levels vary a lot in the subclones. The authors should clarify which subclone was used in the subsequent high-through analyses. And reproducible data (at least by locus-specific analysis) in different subclones would make the conclusions more convincing.

We thank Reviewers for the insightful and very constructive comments, which were helpful in revising and strengthening this manuscript.

New data are shown in Figs.: 3f, 5d, 5e, 6a-g, 7a-e and Suppl. Figs. 2, 4, 5, 7, 8, 10, 11.

Reviewer 1, Comment 1: In the manuscript entitled “ASXL2 links MLL4 and BAP1 on enhancers”, the authors reported the identification of an MLL-binding helix (MBH) domain within ASXL2 protein, which links MLL3/4 to the BAP1 complex. They then further identified a subset of genes that were co-occupied and co-regulated by MLL4 and BAP1.

Major concerns-

In Line 130, “conserved function of MLL3PHD2/3 and MLL4PHD2/3 that can bridge the H3K4-specific methyltransferase MLL3/4 complexes with the H2AK119ub-specific deubiquitinase PR-DUB complex.” The authors need to deplete the HBM motif from the full-length ASXLs, and further evaluate the protein-protein interaction between endogenous MLL3/4 (not just the PHD domains) and wild-type/HBM-depleted ASXLs.

Author’s response: following the reviewer’s suggestion, we generated mouse embryonic stem cells (ESCs) with deletion of the MBH motifs of ASXL1 and ASXL2 (we found that ASXL3 is not expressed in ESCs, please see new Fig. 6a) and investigated the association between endogenous MLL4 and BAP1 in wild-type and MBH-deleted ESCs. Because antibodies against mouse ASXL1/2 or BAP1 are unavailable, we infected cells with lentiviruses expressing doxycycline-inducible BAP1-T7. Immunoprecipitation using an antibody against MLL4 followed by immunoblotting with a T7 antibody revealed that BAP1 co-immunoprecipitates with endogenous MLL4 in wild-type cells, whereas this co-immunoprecipitation was markedly reduced in MBH-deleted cells. These results indicate that MBH of ASXL1/2 is required for the association of MLL4 with BAP1 in ESCs. These new data are shown in Fig. 6b-g.

Furthermore, our data using full-length WT and MBH-deleted ASXL2, purified from K562 cells, demonstrate the association of endogenous MLL3/4 complex with the native PR-DUB complex, and the requirement of ASXL2 MBH for this association (Fig. 5).

Reviewer 1, Comment 2:

In Figure 3F, the authors need to explain why they used leukemia cell line HL60 for the IP experiment between endogenous ASXL2 and MLL3/4 PHD domains? In addition, they need to provide protein levels for input samples to show that equal levels of wild-type/mutant PHDs were expressed in cells. However, in Figure 3K-M, the authors switched to HEK293T cells to study the chromatin occupancy of flag-tagged mutant ASXL1, which contains leukemia-specific truncated mutations. In addition, the MLL3/4 ChIP-seq should be included to determine the impact of mutant ASXL1 on MLL3/4 recruitment.

Author’s response: we have redone the pull-downs and provide input protein levels. The new data are shown in Fig. 3f and Suppl. Figs. 7, 8.

We agree that the data previously displayed in Figure 3k-m did not provide enough information and removed them.

Instead, we performed ChIP-seq assays (new data are shown in Fig. 7) to determine the genomic binding regions of MLL4 and BAP1-T7 and the enrichment of H3K4me1 and H3K27ac in wild-type and MBH-deleted ESCs. In MBH-deleted cells, we observed a significant reduction in localization of BAP1 to active enhancers. In contrast, occupancy of MLL4 on active enhancers remained largely unchanged in MBH-deleted cells.

Reviewer 1, Comment 3: In Figure 3I, the results from the clinical studies were not convincing. The p-value needs to be provided for each of the analysis.

Author's response: we agree and removed Fig. 3i panel.

Reviewer 1, Comment 4: In Fig 4K, the peptide number of MLL3 was 10 times higher than MLL4, suggesting that MLL3/ASXL2 interaction is much stronger than MLL4/ASXL2, which is not in agreement with their NMR results. Also, the subunits RBBP5 and DPY30 were missed in the mass spec results, which are critical for the catalytic activity of MLL3/4. The authors need to conduct the in vitro methylation assay to show that the purified MLL3/4 complex by ASXL2 still has lysine methyltransferase activity.

Author's response: we guessed that the missing subunit in mass spec results the reviewer is referring to was WDR5 instead of RBBP5, as the latter was in the table. WDR5 is a frequent contaminant in affinity purifications from nuclear extracts (CRAPome.org) and did not show a strong decrease of counts like the other MLL3/4 subunits in the MBH-deletion mutant. As for DPY30, its' very small size (11 kDa) makes it often difficult to detect by mas spec. We have now added these data in Fig. 5b and Suppl. Fig. 9.

As for the relative detection of MLL3 and MLL4 in ASXL2 purification, there is no strict direct correlation between MS proteomics total counts and the strength of interactions. This difference may be due to the contact of K562 cells in which expression/abundance/signaling-regulatory events may affect the detected interaction of one versus the other in these specific cells. Binding affinities measured by fluorescence and NMR titration experiments clearly show that either MLL3 or MLL4 PHD2/3 bind to ASXLs similarly, in a 1 μ M range, and based on the high conservation of their sequences and nearly identical binding sites, a similar strength of the interaction is foreseeable.

Importantly, as suggested, we carried out in vitro methylation assay on core histones and nucleosomes using affinity-purified native WT and MBH-deleted ASXL2 complexes. We confirmed the copurification of the MLL3/4 complex, as mono-methylation of H3K4 is detected with the WT fraction but not with the MBH mutant. These results are shown in new Fig. 5e.

Reviewer 1, Comment 5: In Figure 5E, based on the Venn-diagram analysis, only 4.5% (2,843 of 62,590) of BAP1 peaks were co-localized with MLL4. The authors need to conduct a ChIP-seq experiment using T7 antibody in wild-type cells as a negative control (Input or IgG is not be a good control here). In addition, the authors also need to conduct ASXL2 ChIP-seq and compare MLL4 peaks with BAP1/ASXL2 overlapping peaks.

Author's response: Due to the unavailability of antibodies for detecting mouse BAP1 or ASXL1/2, we used the T7 antibody for profiling BAP1 localization. The specificity of the T7 antibody has been previously thoroughly validated in our laboratory using wild-type cells that do not express T7-tagged proteins.

Reviewer 1, Comment 6: In Figure 5F, it seems that BAP1's occupancy is also reduced at BAP1+/MLL4+ common loci in MLL4-KO cells. It is necessary to include H2AK119Ub levels here. My major concern comes from the cell line the authors used for their genome-wide studies (Figure 5-7). It has already been known that there is no detectable MLL3 protein in HCT116 cells. However, at least one copy of MLL4 gene is also mutated (P2442fs and V160M, based on COSMIC and DepMap Data)

in this cell line, but it is still not clear whether these mutations are gain-of-function or loss-of-function mutations. Therefore, HCT116 cell line is not an appropriate model to establish for their NGS studies.

Author's response: we thank this reviewer for pointing to the caveat using HC116 cells and turned to mouse ESCs for conducting genomic analyses (new Figs. 6, 7).

Other suggestions-

In Fig 4J, ASXL2 protein levels should be included. - we apologize for not making it clearer: the Flag signal in the western blot represents ASXL2 protein levels in the affinity-purified fractions, which were analyzed by immunoblot, silver staining, mass spectrometry and used in DUB and HMT assays (please see new Fig. 5a-d and Suppl. Fig. 9). We have added the labeling for clarity.

Reviewer 2, Comment 1: In this comprehensive study, Zhang et al. identified a critical interaction between MLL3/4 and ASXL protein family and demonstrated that such interaction plays an important role in co-localizing MLL3/4 complexes and PR-DUB complex on promoters and enhancers to regulate gene expression. The authors first applied a combination of techniques including phage display, NMR, and fluorescence spectroscopy to discover and characterize a binding interface mediated by MLL3/4 PHD2/3 and ASXL-MBH, and then identified a specific set of genes that are likely regulated by such interaction via CHIP-Seq. The evidences provided are solid and convincing, and overall, the study is of great interest to the field and provides insights into the regulation of MLL3/4 and PR-DUB complexes in human diseases such as cancer.

I have the following minor concerns that should be addressed to improve the manuscript.

1. The authors screened a peptide library which contains 1 million peptides for binding MLL3 PHD2/3 using phage display and two hits were identified (Fig 1C). Are they the only hits? If not, are there other potential proteins that may also interact with MLL3 PHD2/3? It seems like Fig 1C was not explained well in the text or legend.

Author's response: ASXL2/3 were the only top hits with high confidence scores of 62% and 6.5%. The following hits had scores 0.1-0.01%. We have expanded Fig. 1c legend to clarify the data.

Reviewer 2, Comment 2: For NMR-HSQC data, (i) since the residues of MLL3 (or 4) PHD2/3 are assigned, I encourage the authors to provide a chemical shift change (or contour level change) profile for each residue of MLL3 (or 4) PHD2/3 upon ASXL binding. That helps visualize the key residues involved in the binding interface. (ii) For selected HSQC spectra, perhaps arrows could be applied to indicate the shift of the peaks and some key peaks could be labeled with residues if assigned.

Author's response: we assigned only the complex, and the slow exchange regime did not allow showing CSPs per residue or drawing arrows in HSQC panels.

Reviewer 2, Comment 3: I suggest supplementary FigS2 be moved to main Figure 2. Zinc fingers residues and interface residues can be highlighted since they are mentioned throughout the manuscript.

Author's response: because we refer to this alignment prior to Fig. 1j, we would suggest keeping it in a supplement to avoid showing the second alignment in the middle of main Fig. 1 and overwhelm it. As suggested, we have highlighted zinc coordinating and described in the text residues in this figure (it's now Suppl. Fig. 3).

Reviewer 2, Comment 4: MLL3/4 contains several PHD domains, are they similar? Have the authors tested their binding to AXSL peptides? A sequence alignment of the PHD domains especially the interface residues could help elucidate the specificity of PHD2-3/AXSL interaction.

Author's response: MLL4 has seven PHD fingers and MLL3 has eight, and despite the importance of MLL4/MLL3, functions/ligands of these PHDs [with the exception of MLL4_{PHD6} (or homologous MLL3_{PHD7}) that has recently been shown to bind histone H4 selecting for acetylated lysine 16] are essentially unknown.

This is due to the gigantic size of these proteins and the challenge to produce properly folded, non-aggregated PHD fingers, particularly double and triple PHD finger cassettes that contain numerous zinc ion-coordinating and non-coordinating cysteine residues. (Of 28 constructs of MLL4_{PHD456} and MLL4_{PHD45} that we have generated, only two constructs of MLL4_{PHD456} (and none of MLL4_{PHD45}) are expressed as soluble, non-aggregated proteins and therefore suitable for biochemical and structural analysis.)

We were able to produce and tested MLL3_{PHD4} and MLL4_{PHD456}, they do not bind ASXLs.

We have added the following sentence to Fig. 1a legend: "Sequence alignment of all PHD fingers of MLL3 and MLL4 is reported in¹⁰." This citation is a review in which we show sequence alignment of all PHD fingers of MLLs and analyze these sequences (it's a very large figure).

Reviewer 2, Comment 5: For supplementary FigS3, it seems like the contour levels for each spectrum are not normalized. Line 138-140 "The 1H,15N HSQC spectrum of the linked MLL4PHD2/3-ASXL2 construct overlaid well with the spectrum of the ASXL2-bound unlinked MLL4PHD2/3 construct" should be rephrased since they are not exactly overlapped. Key residues could be labeled on the spectra.

Author's response: thank you for catching this, we have normalized the contour levels in this figure (it's now Suppl. Fig. 4). The phrase "overlaid well" has been revised to "overlaid adequately".

Reviewer 2, Comment 6: The authors determined the solution structure by NMR but there is little description regarding the starting template, number of NOEs assigned for structural calculation, and percentage of residue assignments in the text. I also recommend the authors to provide an overlap of structures with top 10 or 15 energy states calculated. Based on the supplementary table 1, the backbone and all heavy atoms RMSDs are estimated to be 1.6 Ang and 1.1 Ang, which are a bit high. Did they include unstructured regions of proteins in the RMSD estimations?

Author's response: we have added NMR structure calculation details in the method section, corrected RMSDs and included new Suppl. Fig. 5 that shows an NMR ensemble of 15 structures.

Reviewer 2, Comment 7: For structural illustration, e.g. Figure 2d and 3a, (i) H-bonds could be labeled using dashed lines to help readers visualize the interactions. (ii) Zinc finger residues could also be displayed since they are mentioned in the text. (iii) It's hard to see how E272K mutation affect the binding based on Fig 3a. (iv) Line 159-160, Residues "652-654" are not labeled in the figure. What kind of interactions does this portion of ASXL2 mediate?

Author's response: (i) as hydrogen bonds cannot be derived from NMR structures, we refer to

short 'within hydrogen bond' distances in the text. (ii) Zinc coordinating residues are now shown as sticks and labeled in Fig. 2a. (iii) E272 forms a negatively charged surface, which can be an essential contact for K661 of ASXL2. (iv) The sentence on Line 159 has been revised to: "The hydrophobic T652-A654 portion of ASXL2 forms a turn and makes van der Waals and hydrophobic contacts..." We have also labeled these residues in Fig. 2d.

Reviewer 2, Comment 8: For figure 3f, the input of GST-tagged MLL3 variants should be provided. Can the authors explain why ASXL2 in pull-down group shows two bands but input group only shows one?

Author's response: we have redone the pull-downs and provide input protein levels. The new data are shown in Fig. 3f and Suppl. Fig. 7, 8.

Reviewer 2, Comment 9: Line 191-194, did the authors test if the C276Y and C294R of MLL3 or G653E of ASXL2 impair the binding? If not, they should rephrase the text (Line 196-198) "Overall, these findings suggest that the impaired interaction between MLL4^{PHD2/3} and ASXL2 could contribute to the development of solid cancers."

Author's response: this sentence (line 196) has been revised to: "Overall, these findings suggest that the altered interaction between MLL4^{PHD2/3} and ASXL2 could play a role in oncogenesis."

Reviewer 2, Comment 10: Does deletion of MBH affect the overall structure/function of AXSL2 besides MLL3 binding?

Author's response: to answer this question, we performed deubiquitinase assays on chromatin using our affinity-purified native WT and MBH-deleted ASXL2 complexes (shown in new Fig. 5d) and in vitro deubiquitinase assays (shown in Suppl. Fig. 10). Our data show no impact of the MBH deletion on BAP1-mediated deubiquitination of H2AK119.

Reviewer 2, Comment 11: Based on the Chip-seq data, some genes are regulated by MLL3 and BAP1 but independent of AXSL2?

Author's response: we have redone genomic studies in ESCs per comments of reviewers 1 and 3 (new data are shown in Figs. 6 and 7).

Reviewer 2, Comment 12: Both MLL3/4 and AXL2 are multi-domain containing proteins that form large complexes with a variety of proteins. It would be helpful for readers to better/quickly understand if the authors could label the residue numbers for all domain in the domain-diagram (Figure 1A and 1M) and also indicate the identified key binding partners in the figure.

Author's response: these are indeed very large proteins, and to avoid diverging the focus from the described interacting regions, we would suggest keeping other regions less labeled with numbers in Fig. 1. We have added the total residue number labels in Fig. 1a and m. The list of subunits known to be part of native ASXL2 and MLL3/4 complexes has been included in the mass spec table (Fig. 5b).

Reviewer 3, Comment 1: Based on careful biochemical characterizations of PHD2/3 fingers of MLL3 and MLL4, this manuscript by Zhang et al. links the COMPASS complex to PR-DUB. In addition to confirming the physical interaction between ASXL proteins with MLL3/4, the authors have demonstrated that the two complexes are interdependent at co-localized enhancers and may co-regulate the transcription of a subset of target genes. This study has provided a clear structural basis

for the interaction between the two complexes. But quite a few concerns discourages us from recommendation of its publication in the current form.

Major comments:

1. There lacks genomic and functional analyses of MBH. The identification of MBH is the most interesting finding in this study. However, the authors did not center this domain to design experiments. First, MBH's necessity seems different in solid cancers and hematopoietic malignancies. But the authors did not even discuss about the underlying cause for these differences. Second, the only experimental data showing the importance of MBH are shown in Figure 3k-m. These data are generally weak and not that convincing. Besides, the data show that ASXL1 is localized at TSS in a MBH-dependent manner, which is inconsistent with the dominant colocalization of COMPASS complex and PR-DUB at enhancers. Third, no rescue experiments in ASXL2KO cells with either WT or MBH deletion/mutation were designed. The CHIP-seq analyses even functional analyses with these rescues are critical to claim the importance of MBH.

Author's response: we appreciate the reviewer's comments that this study 'has provided a clear structural basis for the interaction between the two complexes'. We note that despite the high importance of MLL4/MLL3, functions of their seven/eight PHD fingers (except for one), are essentially unknown. Progress in this direction remains very slow (please see our response to Comment 4 of Rev 2).

We agree that the data previously displayed in Figure 3i-m are generally weak and removed them.

We also agree that HCT116 is not a suitable model system for our genomic study due to MLL4 mutations, therefore, we turned to mouse embryonic stem cells (ESCs). We have generated mouse ESCs with the deletion of the MBH motifs in ASXL1 and ASXL2 (we found that ASXL3 is not expressed in ESCs, please see new Fig. 6a) and investigated the association between endogenous MLL4 and BAP1 in wild-type and MBH-deleted ESCs. Because antibodies against mouse ASXL1/2 or BAP1 are unavailable, we infected cells with lentiviruses expressing doxycycline-inducible BAP1-T7. Immunoprecipitation using an antibody against MLL4 followed by immunoblotting with a T7 antibody revealed that BAP1 co-immunoprecipitates with endogenous MLL4 in wild-type cells, whereas this co-immunoprecipitation was markedly reduced in MBH-deleted cells. These results indicate that MBH of ASXL1/2 is required for the association of MLL4 with BAP1 in ESCs. These new data are shown in Fig. 6b-g. Our CHIP-seq data clearly support the significance of MBH in facilitating BAP1 binding to MLL4 bound active enhancers (new Fig. 7).

Reviewer 3, Comment 2: We are highly concerned about the BAP1-T7/MLL4 CHIP-seq data, as shown in Figures 5-7. The TPM/FPKM values and genomic localizations should be labelled on all the tracks. The comparison should be group auto-scaled. However seeing from the tracks and accessible bigwig files, the BAP1-T7/MLL4 peaks do not look right. The sharp peaks without any genomic background are very likely due to a failed library preparation. In this condition, the difference between control and KO groups could be random. The authors should do a careful QC for the original data. And MLL4 CHIP-seq in MLL4KO cells to confirm the signal specificity would be helpful.

Author's response: We have incorporated labeled values onto the genome tracks as suggested. Regarding the absence of genomic background signals, please note that the provided wiggle format underwent additional processing using the SICER algorithm (Zang, Schones et al. 2009). The resulting output files were refined to eliminate background signals

outside of peak regions, ensuring a more focused analysis of the relevant genomic features. If desired, we are also able to provide the raw wiggle files upon request.

Reviewer 3, Comment 3: In addition to the uncertain data quality, the data analyses were poorly performed. Here are a few examples. 1) As we know, ASXL1/2 mutations are frequently found in various subtypes of myeloid malignancy with totally different prognosis. However in Figure 3i, the authors did not distinguish the subtypes and failed to observe the significant difference in prognosis for ASXL1 truncation mutants with MBH or not. The authors are encouraged to make a careful analysis within each subtype of malignancies. 2) In Figures 5 and 6, the authors only show the positive data supporting the interdependency at MLL4/BAP1 co-occupied regions, without convincing negative controls (BAP1 or MLL4-negative regions). Actually looking from the tracks, MLL4 signals are also decreased or lost from BAP1-negative regions. It is very likely that these changes are just random, or secondary to cell fate changes or due to alternative mechanisms. To have a clear conclusion in here, optimal data quality and the rescue experiments as mentioned above are required.

Author's response: as discussed above, in the revised manuscript, we use mouse ESCs as the model system. New ChIP-seq data reveal that the deletion of ASXL1/2 MBH leads to a reduction in BAP1-T7 binding specifically on active enhancers. The ASXL1/2 MBH deletion does not reduce genomic binding of MLL4.

We also removed the data displayed in original Figure 3i-m.

Reviewer 3, Comment 4: MLL3/4 and PR-DUB complexes are well known for their biochemical activities on histones. To establish the functional links, the authors should show the effects on H3K4me1 and H2AK119Ub1 at the genome wide in either MLL4 or PR-DUB deficient cells. Without these data, we can hardly understand the biological significance for the two coupled complexes in enhancer regulation.

Author's response: We performed ChIP-seq analysis of H3K4me1 in wild-type and MBH-deleted ESCs. The deletion of ASXL1/2 MBH had no impact on the enrichment of H3K4me1 on enhancers, aligning with the absence of effect on the genomic binding of MLL4 (new Fig. 7).

Furthermore, the data shown in new Fig. 5d, e demonstrate that affinity-purified PR-DUB/ASXL2 has both H2AK119 deubiquitinase and H3K4 monomethyltransferase activities, and the latter one depends on the association of the MLL3/4 complex through the ASXL2 MBH domain.

Reviewer 3, Comment 5: Are there any effects on cell growth after deletion of BAP1 or MLL4 in HCT116 cells? BAP1 deletion usually leads to cell growth defects, which possibly makes the subclones different. Looking from Figure 5d, H3K4me1 levels vary a lot in the subclones. The authors should clarify which subclone was used in the subsequent high-through analyses. And reproducible data (at least by locus-specific analysis) in different subclones would make the conclusions more convincing.

Author's response: In the MBH-deleted ESCs, we did not observe any noticeable growth defects. The following sentence has been added on page 12: "No noticeable changes in ESC identity transcription factor Oct4 levels or cell morphology were observed between WT and Δ MBH-1/2 cells (Fig. 6e, f)."

REVIEWER COMMENTS

Reviewer #1 (Remarks to the Author):

In the revised manuscript, Zhang et al. removed most of the unconvincing data from the original submission and provided new results to claim the functions of the MBH domain of ASXLs. Regrettably, the recent experiments carried out in mouse ES cells do not substantiate the proposed conclusion. Here are my major concerns-

1. The BAP1-T7 inducible expressing system is confusing. The author induced BAP1-T7 expression in mouse ES cells for a couple of days and conducted all the biological experiments with these cells. However, it is crucial to note that the functionality of the BAP1 enzyme relies on its incorporation into a multiprotein complex at the chromatin. The overexpressed BAP1 with T7 tag needs to compete with the endogenous BAP1 for the other essential subunits (ASXLs, FOXKs, HCFC1....). Importantly, the surplus BAP1 lacking subunits will stay in the cytoplasm and function as a random deubiquitination enzyme. It is difficult to understand why the authors conducted all the biological experiments without detecting and showing protein levels of endogenous BAP1, ASXL1, or ASXL2, even though there are commercial antibodies for all of them.

2. In Figure 6, the authors deleted the MBH domain from ASXL1 and ASXL2 from mouse ES cells. However, there is no evidence to show that the protein levels of the mutant ASXL1/ASXL2 are still expressed at the same levels as wild-type ASXL1/2 in cells.

3. In Figure 7, since mouse ES cells express both MLL3 and MLL4, the authors must provide MLL3 recruitment data at genome-wide levels. It is also difficult to understand why loss of BAP1 leads to an increase of MLL4 occupancy at the active enhancer regions (Fig. 7c). The BAP1 occupancy at other regions (promoter, primed enhancer...) should also be shown between WT, delta-MBH-1/2 cells. In addition, the authors need to compare the H2AK119ub levels between WT, delta-MBH-1, and delta-MBH-2 cells.

Reviewer #2 (Remarks to the Author):

The revised manuscript has been significantly improved. I feel that all my concerns have been properly addressed.

Reviewer #3 (Remarks to the Author):

The authors have used totally different cellular systems and addressed most of my concerns. A few uncertainties are about the effects of ASXL1-MBH deletion on MLL4 binding and H2AK119ub1 levels in vivo. Though it is convincing that the MBH is dispensable for H2AK119 deubiquitylation activity, we are still curious whether the decreased binding of BAP1 may affect H2AK119ub1 deposition at chromatin, given that BAP1 binding seems markedly decreased. Reviewer 1 also raised this issue but the authors did not respond. And the authors did not explain why MLL4 at active enhancers remain unchanged when the interaction between the two complexes is compromised. According to the current data in Figure 7, we are still confused about the biological significance of coupling these two complexes.

We thank Reviewers for the insightful and very constructive comments, which were helpful in revising and strengthening this manuscript.

Reviewer #1 (Remarks to the Author)

In the revised manuscript, Zhang et al. removed most of the unconvincing data from the original submission and provided new results to claim the functions of the MBH domain of ASXLs. Regrettably, the recent experiments carried out in mouse ES cells do not substantiate the proposed conclusion. Here are my major concerns-

1. The BAP1-T7 inducible expressing system is confusing. The author induced BAP1-T7 expression in mouse ES cells for a couple of days and conducted all the biological experiments with these cells.

Because we use mouse ESCs and no commercial BAP1 antibodies are available that are suitable for IP or ChIP in mouse cell lines, we have to express BAP1-T7 to perform these experiments.

However, it is crucial to note that the functionality of the BAP1 enzyme relies on its incorporation into a multiprotein complex at the chromatin. The overexpressed BAP1 with T7 tag needs to compete with the endogenous BAP1 for the other essential subunits (ASXLs, FOXKs, HCFC1....).

BAP1 has previously been shown to fully integrate into complexes when overexpressed using retroviral systems and is functional (see PMID: 20805357). In support, our data confirm that BAP1-T7 is integrated into the complex and pulls down endogenous ASXL1 and FOXK1 subunits in ESCs (new Fig. 6g).

Importantly, the surplus BAP1 lacking subunits will stay in the cytoplasm and function as a random deubiquitination enzyme.

Cytosolic BAP1 was found to be improperly folded and non-functional. It requires the DEUBAD domain of ASXLs for deubiquitinase activity (see PMID: 26416890; 30349006; 26739236; 24703950).

It is difficult to understand why the authors conducted all the biological experiments without detecting and showing protein levels of endogenous BAP1, ASXL1, or ASXL2, even though there are commercial antibodies for all of them.

As suggested, we have carried out WB to show that the MBH deletion does not affect ASXL1 or ASXL2 protein levels in mouse ESCs (new Fig. 6e). Expression levels of other PR-DUB complex subunits, including BAP1, FOXK1, HCF1 and OGT are also comparable between WT and Δ MBH-1/2 cell lines.

2. In Figure 6, the authors deleted the MBH domain from ASXL1 and ASXL2 from mouse ES cells. However, there is no evidence to show that the protein levels of the mutant ASXL1/ASXL2 are still expressed at the same levels as wild-type ASXL1/2 in cells.

We have carried out WB to show that the MBH deletion does not affect ASXL1 or ASXL2 protein levels (new Fig. 6e).

3. In Figure 7, since mouse ES cells express both MLL3 and MLL4, the authors must provide MLL3 recruitment data at genome-wide levels. It is also difficult to understand why loss of BAP1 leads to an increase of MLL4 occupancy at the active enhancer regions (Fig. 7c). The BAP1 occupancy at other regions (promoter, primed enhancer....) should also be shown between WT, delta-MBH-1/2 cells.

We and others have previously reported that Mll4 but not Mll3 plays a pivotal role in mouse ES cells and during mouse early embryo development (PMID: 24368734; 27698142; 32439762; 37012455). As shown in new Suppl. Fig. 14, in mouse ES cells, the mRNA level of Mll4 is 3-fold higher than the mRNA level of Mll3. These findings prompted us to focus on MLL4 in mouse ES cells.

We appreciate the reviewer's point regarding MLL3- we agree and revised the title (removed MLL3): "ASXLs binding to the PHD2/3 fingers of MLL4 provides a mechanism for the recruitment of BAP1 to active enhancers."

We also appreciate the comment regarding a mildly increased MLL4 occupancy in average profiles, which was due to a few extremely high outliers. Heat maps do not show such an increase (Fig. 7c and new Fig. 7d).

The data on BAP1 occupancy at promoters, primed enhancers, active enhancers and other regions is shown in new Suppl. Fig. 16.

In addition, the authors need to compare the H2AK119ub levels between WT, delta-MBH-1, and delta-MBH-2 cells.

As suggested, we have performed ChIP-qPCR in ASXL2 KD K562 cells complemented with WT ASXL2 or delta-MBH mutant ASXL2 (new Fig. 5f, g). We found that loss of the interaction of ASXL2 with MLL3/4 leads to an increase of H2AK119ub at an active enhancer, while H3K4me1 level is not affected. These new data agree with the results obtained in mESCs and support the idea that MLL3/4 has a role in recruiting BAP1 at active enhancers.

We believe that time and labor consuming ChIP-seq of H2AK119ub in mouse ESCs (delta-MBH-1 and 2) will not provide a substantially new mechanistic information, which is the focus of this work.

Reviewer #2 (Remarks to the Author)

The revised manuscript has been significantly improved. I feel that all my concerns have been properly addressed.

Reviewer #3 (Remarks to the Author)

The authors have used totally different cellular systems and addressed most of my concerns. A few uncertainties are about the effects of ASXL1-MBH deletion on MLL4 binding and H2AK119ub1 levels in vivo. Though it is convincing that the MBH is dispensable for H2AK119 deubiquitylation activity, we are still curious whether the decreased binding of BAP1 may affect H2AK119ub1 deposition at chromatin, given that BAP1 binding seems markedly decreased. Reviewer 1 also raised this issue but the authors did not respond. And the authors did not explain why MLL4 at active enhancers remain unchanged when the interaction between the two complexes is compromised. According to the current data in Figure 7, we are still confused about the biological significance of coupling these two complexes.

We thank this reviewer for the valuable comments and suggestions. Our new data, including measurements of H2AK119ub in ASXL2 KD K562 cells complemented with WT ASXL2 or delta-MBH mutant ASXL2 (new Fig. 5f, g) show that loss of the interaction of ASXL2 with MLL3/4 leads to an increase of H2AK119ub at an active enhancer, while H3K4me1 level is not affected. These new data agree with the results obtained in mESCs and support the idea that MLL3/4 has a role in recruiting BAP1 at active enhancers.

Regarding the significance of our work- our study reports the discovery and characterization of the molecular mechanism underlying the interaction between PHD fingers of MLL3/4 and the MBH region of ASXLs. Identifying the atomic-resolution molecular interface between these proteins is a significant breakthrough that could help to design molecules to mediate/manipulate the relationship between these essential proteins, MLL3/4 and ASXLs.

Regarding the unchanged MLL4 binding at active enhancers, we speculate that MLL4 is directly recruited by transcription factors to active enhancers. Our data suggest that binding of BAP1 to active enhancers is facilitated by MLL4, but not vice versa.

REVIEWERS' COMMENTS

Reviewer #3 (Remarks to the Author):

The authors have addressed my concerns. Just one thing. The authors explain in the response that the binding of BAP1 to active enhancers is facilitated by MLL4, but not vice versa. Notably, this conclusion is contrary to Shilatifard's paper in Nat Med. The authors should at least discuss about this discrepancy.

Please find attached the revised manuscript entitled “ASXLs binding to the PHD2/3 fingers of MLL4 provides a mechanism for the recruitment of BAP1 to active enhancers”, submitted for your consideration. The manuscript has been revised to address the editorial comments and the remaining Reviewer’s comment.

Reviewer #3 (Remarks to the Author):

The authors have addressed my concerns. Just one thing. The authors explain in the response that the binding of BAP1 to active enhancers is facilitated by MLL4, but not vice versa. Notably, this conclusion is contrary to Shilatifard's paper in Nat Med. The authors should at least discuss about this discrepancy.

We cite the Nature Med paper (ref. #16). The differences might be due to different cell lines (HEK293T, MDA-MB-453 and MDA-MB-231 were used in ref. #16).

I thank you for your consideration and look forward to your reply.

Sincerely,

Tatiana Kutateladze
Professor
Department of Pharmacology
University of Colorado School of Medicine